# ON ENTROPY CONTROL IN LLM-RL ALGORITHMS

**Han Shen**
Ant Group
shenhanhs@gmail.com

## ABSTRACT

For RL algorithms, appropriate entropy control is crucial to their effectiveness. To control the policy entropy, a commonly used method is entropy regularization, which is adopted in various popular RL algorithms including PPO, SAC and A3C. Although entropy regularization proves effective in robotic and games RL conventionally, studies found that it gives weak to no gains in LLM-RL training. In this work, we study the issues of entropy bonus in LLM-RL setting. Specifically, we first argue that the conventional entropy regularization suffers from the LLM's extremely large response space and the sparsity of the optimal outputs. As a remedy, we propose AEnt, an entropy control method that utilizes a new clamped entropy bonus with an automatically adjusted coefficient. The clamped entropy is evaluated with the re-normalized policy defined on certain smaller token space, which encourages exploration within a more compact response set. In addition, the algorithm automatically adjusts entropy coefficient according to the clamped entropy value, effectively controlling the entropy-induced bias while leveraging the entropy's benefits. AEnt is tested in math-reasoning tasks under different base models and datasets, and it is observed that AEnt outperforms the baselines consistently across multiple benchmarks.

## 1 INTRODUCTION

RL seeks to maximize the reward received by a sequential decision making system. In recent years, RL has proven to be an effective tool for training LLMs (Yang et al., 2025; DeepSeek-AI, 2025; Comanici et al., 2025). The advances of LLMs in math, coding and planning tasks has been astonishing, with their performance on competitive benchmarks drastically increasing after RL training.

The methods used in LLM-RL are predominantly policy-gradient based, e.g., the PPO (Schulman et al., 2017) family. Policy gradient based methods reinforce the sampled actions that lead to higher rewards compared to other sampled actions. However, when the optimal actions are not sampled, the policy gradient methods can over-reinforce the sampled locally optimal actions, ultimately resulting in the policy stuck at suboptimal points (Agarwal et al., 2021). The sub-optimal actions can be meaningless in deep RL and oftentimes have a large performance gap from the optimal ones (Henderson et al., 2018), e.g., in LLM tasks, the policy can be stuck at producing the correct format but incorrect results. A straightforward remedy for the issue was the so-called *entropy-regularized RL* methods (Williams & Peng, 1991), where the policy maximizes a sum of rewards and some *entropy bonus* (policy randomness). This technique was commonly used in policy-gradient methods including A3C (Mnih et al., 2016), PPO (Schulman et al., 2017) and SAC (Haarnoja et al., 2018), providing strong benefits in tasks requiring hierarchical behaviors. Intuitively, the entropy bonus keeps the policy random and explorative, thus prevents the policy from over-reinforcing certain actions and getting stuck. Moreover, entropy regularization is shown to provide strong optimization benefits both empirically (Ahmed et al., 2019) and theoretically (Mei et al., 2020; Klein et al., 2023).

However, it is observed that entropy regularization offers little gains in LLM-RL training. Specifically, the experimental results to be shown in Section 5 suggest that entropy-regularized GRPO yields minimal gain compared to basic GRPO. In addition, Cui et al. (2025) observes that the validation accuracy is unchanged under different scaling of the entropy bonus in some LLM-math tasks. These results are particularly underwhelming compared to those in other deep RL tasks including robotics and games, where the benefit of entropy bonus is significant (see, e.g., (Haarnoja et al., 2018, Figure 3)). Moreover, such empirical contradiction also indicates a theoretical gap between the existing

analysis which justifies the entropy's benefit (Mei et al., 2020) and its effect in LLM-RL. Therefore, a careful study and a remedy for this issue is in dire needs, as the potential gain from entropy bonus is yet to be unlocked for LLM training.

In this work, we first give a theoretical view of the entropy effect in LLM-RL training, which explains the conventional entropy bonus's emergent issues in LLM tasks. To that end, we then propose AEnt, an entropy regularization method that uses an adaptive and clamped entropy bonus. Our main contribution is twofold:

- **A theory on the entropy effect and its issues in LLM-RL.** Under no entropy bonus, we show that entropy collapse indicates learning stagnancy and give a performance bound. Then we show that entropy regularization can fail to improve this result under LLM's large response space and the task's sparse optimality.

- **AEnt, a recipe to enable effective entropy regularization.** Inspired by the theoretical analysis, we then propose a recipe for this issue. Instead of using the traditional entropy bonus, AEnt uses a clamped entropy defined with the re-normalized LLM policy on a size-reduced token space. The clamped entropy only smooths out policy on the reasonable responses set, which enjoys decreased bias compared to the original entropy. Furthermore, the clamped entropy bonus is scaled with a coefficient that gets automatically adjusted to balance its bias and benefits. Empirical evidence suggests that AEnt consistently improves over the baselines across multiple benchmarks.

## 1.1 RELATED WORKS

**Policy-gradient based LLM-RL algorithms.** The RL algorithms used in LLM post-training have been predominantly policy-gradient based (Sutton et al., 1999). They are either based on PPO (Schulman et al., 2017) (see, e.g., GRPO (Shao et al., 2024), DAPO (Yu et al., 2025) and (Fu et al., 2025)), or the more basic REINFORCE algorithms (Williams, 1992) (see, e.g., (Ahmadian et al., 2024; Chu et al., 2025)). Though PPO was initially proposed in actor-critic style, the critic is replaced with Monte-Carlo rollout in resource-limited or outcome-driven LLM training scenarios. Contrary to the practice in robotic and games RL (Mnih et al., 2016; Schulman et al., 2017), the fore-mentioned LLM-RL algorithms do not consider entropy regularization.

**Entropy regularization in RL.** Entropy-regularized RL was initially introduced in (Williams & Peng, 1991). It has been commonly used in various popular policy-based deep RL algorithms (Mnih et al., 2016; Schulman et al., 2017; Haarnoja et al., 2018) which have provided ample empirical evidence for its effectiveness in robotic and games tasks. Entropy regularization's optimization benefits have also been empirically (Ahmed et al., 2019) and theoretically (Mei et al., 2020; Klein et al., 2023) studied. However, it does not give notable performance gains for LLMs (see, e.g., Section 5 and (Cui et al., 2025)). As a result, alternative entropy control techniques are often adopted. In (Zhang et al., 2024) reshapes the reward function to regulate the policy. Or in a concurrent work (Cui et al., 2025), the algorithm clips or regulates the parts the policy update that decrease entropy too much. To our best knowledge, existing works do not answer the question of why and when entropy regularization can fail in LLM-RL, and have not uncovered the potential benefits of entropy bonus.

## 2 PRELIMINARIES

In this section, we will first give formal definitions of some RL concepts, and then introduce several prominent policy optimization algorithms.

**Finite-horizon Markov decision process (MDP).** In LLM-RL setting, the learning task can be modeled as a finite-horizon MDP defined by a $\mathcal{M} = \{\mathcal{S}, \mathcal{A}, \mathcal{P}, r, H\}$, where $\mathcal{S}$ is a finite state space (e.g., input token sequence of the LLM), $\mathcal{A}$ is a finite action space (e.g., LLM's vocabulary), and the state transits by $s_{t+1} = \mathcal{P}(s_t, a_t)$ where $\mathcal{P}$ is a concatenation operation of the input sequence $s_t$ and the output token $a_t$. Function $r(s, a) \in [0, 1]$ assigns a reward to $(s, a)$. Horizon $H$ is the max response length. An LLM-policy parameterized by $\theta \in \mathbb{R}^d$ is denoted as $\pi_\theta(a|s)$, which assigns a probability for each token $a \in \mathcal{A}$ given input $s \in \mathcal{S}$.

**RL objectives.** Given the initial time step $h$ and state $s_h = s$, define the cumulative reward as

$$V_h^{\pi_\theta}(s) := \mathbb{E}_{\pi_\theta}\Big[ \sum_{t=h}^{H-1} r(s_t, a_t)|s_h = s\Big] \tag{2.1}$$

where $\pi_\theta(s_t) := \pi_\theta(\cdot|s_t)$, the expectation is taken over the trajectory $(a_t, s_{t+1}, \ldots, a_{H-1})$ where $a_t \sim \pi_\theta(s_t)$ for each $t$. Given a dataset $\mathcal{D}$ containing input queries, the objective of RL is

$$\max_\theta V^{\pi_\theta}(\mathcal{D}) := \mathbb{E}_{s \sim \mathcal{D}}[V^{\pi_\theta}(s)] = \mathbb{E}_{\pi_\theta}\Big[ \sum_{t=0}^{H-1} r(s_t, a_t)\Big] \tag{2.2}$$

where $V^{\pi_\theta}(s) = V_0^{\pi_\theta}(s)$, and we omit time step subscripts for the value functions of step 0.

**Entropy regularized RL.** Given $\mathcal{D}$, we can define the entropy of the policy $\pi_\theta$ as

$$\mathcal{H}(\pi_\theta) := -\mathbb{E}_{\pi_\theta}\Big[ \sum_{t=0}^{H-1} \log \pi_\theta(a_t|s_t)\Big] \tag{2.3}$$

Entropy regularized RL aims to maximize the entropy regularized objective $V_\lambda^{\pi_\theta}(\mathcal{D}) := V^{\pi_\theta}(\mathcal{D}) + \lambda \mathcal{H}(\pi_\theta)$. Due to space limitation, we defer some definitions to Appendix A.1.

**PPO-clip family.** To solve for 2.2, a prominent algorithm is the PPO-clip (Schulman et al., 2017). Given the sampling policy $\pi_b$, the objectives of PPO-clip algorithms can be written as

$$\mathcal{L}_{\mathrm{PO}}(\theta) = \mathbb{E}_{s_0 \sim \mathcal{D}, \{a_t \sim \pi_b(s_t)\}_{t \leq H-1}}\Big[ \min\Big(\frac{\pi_\theta(a_t|s_t)}{\pi_b(a_t|s_t)}\hat{A}_t, \mathrm{clip}\Big(\frac{\pi_\theta(a_t|s_t)}{\pi_b(a_t|s_t)}, 1-\epsilon_{\mathrm{low}}, 1+\epsilon_{\mathrm{high}}\Big)\hat{A}_t\Big)\Big] \tag{2.4}$$

where $\hat{A}_t$ is an estimate of the advantage function. GRPO uses a Monte-Carlo estimate of the trajectory-level advantage. DAPO additionally decouples the clip ratio by setting different $\epsilon_{\mathrm{low}}, \epsilon_{\mathrm{high}}$ and incorporates extra sampling constraints and overlong response penalty. Given a suitable clip range, the PPO-clip algorithm can be viewed as a policy gradient algorithm (Jin et al., 2023).

## 3 A THEORY ON ENTROPY EFFECT IN POLICY GRADIENT BASED LLM-RL

In this section, we give some theoretical insights into LLM-RL training. We will show performance bounds for RL algorithms without entropy control or with conventional entropy control. We will also draw connections with some concurrent works based on our theoretical insights.

Suppose the LLM is a softmax policy, that is

$$\pi_\theta(a|s) = \frac{\exp(\theta_{s,a})}{\sum_a \exp(\theta_{s,a})}$$

where $\theta_{s,a}$ is the logit of token $a$ given input $s$. The policy gradient based algorithms without entropy regularization, including PPO-clip, are generally guaranteed to converge to an $\epsilon$-stationary point of the RL objective $V^{\pi_\theta}(\mathcal{D})$ satisfying $\|\nabla V^{\pi_\theta}(\mathcal{D})\| \leq \epsilon$ (Agarwal et al., 2021; Jin et al., 2023). When doing policy optimization without regularization, (Cui et al., 2025) observes that the policy entropy quickly diminishes as performance increases, and ultimately performance saturates when entropy completely collapses. In the following result, we give some theoretical insights into this observation.

**Proposition 1** (Bounds under no entropy control). *Assume the policy is a softmax. We have:*

(I) *Policy entropy is an upper bound of the policy gradient:* $\big\|\nabla V^{\pi_\theta}(\mathcal{D})\big\| \leq 2\mathcal{H}(\pi_\theta)$.

(II) *If* $\big\|\nabla V^{\pi_\theta}(\mathcal{D})\big\| \leq \epsilon$, *then given any query* $s_0$ *in dataset* $\mathcal{D}$, *the policy suboptimality on the query satisfies*

$$V^{\pi^*}(s_0) - V^{\pi_\theta}(s_0) \leq \frac{1}{C^{\pi_\theta}(s_0)}\epsilon \tag{3.1}$$

*where* $\pi^* \in \arg\max_\pi V^\pi(\mathcal{D})$, $C^{\pi_\theta}(s_0) := \frac{1}{\sqrt{H|\mathcal{D}|}}\max_{(a_0,\ldots,a_{H-1})\in\mathcal{A}_H^*(s_0)} \Pi_{t=0}^{H-1}\pi_\theta(a_t|s_t)$ *in which* $\mathcal{A}_H^*(s_0) = \{(a_0, a_1, \ldots, a_{H-1}) \in \mathcal{A}^H \mid \exists\pi^*, \Pi_{t=0}^{H-1}\pi^*(a_t|s_t) > 0\}$ *is the set of all optimal responses given query* $s_0$.

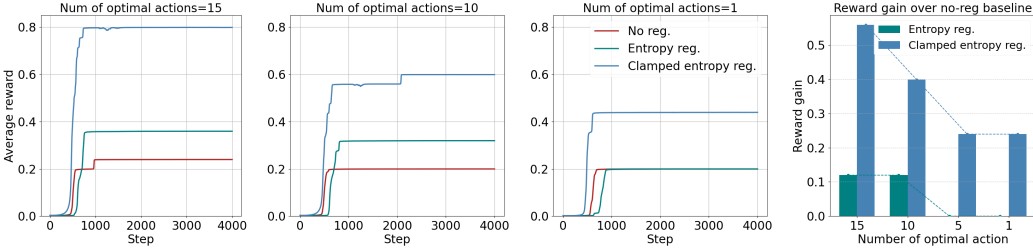

Figure 1: Test in a controlled MDP with a large action space of size $|\mathcal{A}| = 10^5$ and increasingly sparse optimal actions.

The first bullet (I) suggests the policy entropy is an indicator of the policy stationarity, that is, a small entropy indicates a small policy gradient $\left\|\nabla V^{\pi_\theta}(\mathcal{D})\right\|$ and the convergence of the policy. The second bullet (II) quantifies the actual performance of the almost stationary policy, where the reward optimality gap on query $s_0$ is bounded by $\mathcal{O}(\epsilon/C^{\pi_\theta}(s_0))$. The factor $C^{\pi_\theta}(s_0)$ can be controlled (bounded away from $0$) when the initial LLM and the RL algorithm can sufficiently explore the optimal response to $s_0$. For example, one can either use a large batch size (Klein et al., 2023) or a strong initial model (Weissmann et al., 2024) to control $C^{\pi_\theta}(s_0)$. In this case, the performance is ultimately bounded by $\mathcal{O}(\epsilon)$. The error $\epsilon$ decreases with prolonged RL training, while it usually cannot decrease to $0$ due to the presence of sampling noise or the advantage estimation error.

On the other hand, the maximum entropy RL optimizes the entropy-regularized reward sum $V_\lambda^{\pi_\theta}(\mathcal{D})$ (Williams & Peng, 1991). In non-LLM deep RL tasks, this method has long been popular and can significantly outperform methods without entropy control (Mnih et al., 2016; Haarnoja et al., 2018). However, experiments (to be shown in Section 5) show that traditional entropy regularization gives weak to no gains in LLM-RL training. In the next result, we give theoretical insight into this issue.

**Proposition 2** (Bound for entropy-regularized methods). *Assume the policy is a softmax. If* $\left\|\nabla V_\lambda^{\pi_\theta}(\mathcal{D})\right\| \leq \epsilon$, *then given any query $s_0$, the policy suboptimality on the query satisfies*

$$V^{\pi^*}(s_0) - V^{\pi_\theta}(s_0) \leq \frac{1}{C_\lambda^{\pi_\theta}(s_0)} \frac{\epsilon^2}{2\lambda} + \lambda H \log \frac{|\mathcal{A}|}{|\mathcal{A}_H^*(s_0)|^{\frac{1}{H}}} \tag{3.2}$$

*where $C_\lambda^{\pi_\theta}(s_0)$ will be specified in the proof.*

Similar conditions to Propositions 1.(II)&2 have been derived in (Mei et al., 2020) for the discounted infinite horizon MDPs, while our results hold for the finite horizon MDPs under a concatenation transition. Proposition 2 also provides a more accurate bound for the entropy bias.

**Entropy regularization suffers from immense response space with sparse optimality in LLM tasks.** As compared to no entropy control case in Proposition (II), the above bound's dependence on $\epsilon$ improves to $\mathcal{O}(\epsilon^2/2\lambda)$ when $C_\lambda^{\pi_\theta}(s_0)$ is bounded away from $0$. However, this optimization benefit does not come free as a bias term is introduced. The entropy bias is $\mathcal{O}(H \log(|\mathcal{A}|/|\mathcal{A}_H^*(s_0)|^{\frac{1}{H}}))$, which increases with the response space size $H \log |\mathcal{A}|$ and the sparsity of optimal responses $\log(1/|\mathcal{A}_H^*(s_0)|)$. The bias is especially ubiquitous in LLM-RL, where the response space is typically extremely large (hundreds of thousands tokens to sample from in each step) as compared to that in, e.g., classic control and games where the action space size and horizon are typically at the hundreds (Brockman et al., 2016; Silver et al., 2017). To better demonstrate this effect, we report a numerical test result in Figure 1. The test is done in a synthetic MDP with $|\mathcal{A}| = 10^5$, and the optimal action is sparse with various numbers in $\{1, 5, 10, 15\}$ (see Appendix A.5 for details). It is observed that entropy regularization leads to gains over no-regularization when number of optimal actions is $10, 15$, but offers no gains when the optimal action becomes too sparse with fewer than $5$ optimal actions.

To unlock the benefit of entropy regularization, it is crucial to mitigate the negative effect caused by the large response space in LLM tasks. In the following sections, we propose our recipe for this issue and empirically demonstrate its effectiveness.

# 4 AEnt: Adaptive entropy regularization with token space clamping

In this section, we will first introduce the core components of our method and then the full algorithm.

## 4.1 Entropy with token space clamping

Recall the entropy regularized RL objective is $V^{\pi_\theta}(\mathcal{D}) + \lambda\mathcal{H}(\pi_\theta)$ where

$$\mathcal{H}(\pi_\theta) = -\sum_{t=0}^{H-1} \mathbb{E}_{s_t \sim \pi_\theta}\Big[ \sum_{a \in \mathcal{A}} \pi_\theta(a|s_t) \log \pi_\theta(a|s_t)\Big]. \qquad \text{(Entropy)}$$

Entropy is maximized by the uniform policy $\pi_{\text{uniform}}(a|s) = 1/|\mathcal{A}|$. Maximizing the entropy pulls the LLM policy towards $\pi_{\text{uniform}}(a|s) = 1/|\mathcal{A}|$, increasing the likelihood of low-probability actions while decreasing those of the high-probability ones. Intuitively, this helps when the optimal actions have low probabilities and are thus less likely to be sampled and reinforced. Such mechanism works well in the RL tasks where the discrete action space is small (Brockman et al., 2016). While it is extremely inefficient in LLM-RL setting since $\mathcal{A}$ is prohibitively immense with sparse optimal tokens. Specifically, when $1/|\mathcal{A}|$ is small, pulling $\pi_\theta(a|s)$ for every $a \in \mathcal{A}$ towards $1/|\mathcal{A}|$ gives weak gains and produces large bias due to the large amount of non-optimal tokens in the complete token space.

To overcome this issue, we instead use a clamped entropy:

$$\tilde{\mathcal{H}}(\pi_\theta) := -\sum_{t=0}^{H-1} \mathbb{E}_{s_t \sim \pi_b}\Big[ \sum_{a \in \mathcal{A}(s_t)} \tilde{\pi}_\theta(a|s_t) \log \tilde{\pi}_\theta(a|s_t)\Big] \qquad \text{(Clamped entropy)}$$

$$\text{with } \tilde{\pi}_\theta(a|s) = \frac{\exp\big(\theta_{s,a}\big)}{\sum_{a \in \mathcal{A}(s)} \exp\big(\theta_{s,a}\big)} \text{ and } \mathcal{A}(s) = \{\text{top } (1-p) \text{ percent tokens in } \pi_\theta(\cdot|s)\}$$

The clamped entropy is evaluated by a re-normalized policy $\tilde{\pi}_\theta$ on a size-reduced, input-dependent token space $\mathcal{A}(s)$. By the insights from Proposition 2 and Figure 1, regularizing on a smaller response space with denser optimality generally leads to reduced bias and larger gains. With this principle, we set $\mathcal{A}(s)$ as the the top-probability tokens set of $\pi_\theta(s)$. The intuition is that since the base models are pre-trained or fine-tuned prior to the RL phase, the bottom probability tokens are unlikely to be optimal. We find leaving them out reduces entropy-induced bias and leads to performance gains. It can be observed from the toy demonstration in Figure 1 that clamped entropy regularization leads to performance gains when entropy regularization does not (number of optimal actions $\leq 5$), and is generally more robust to optimality sparsity increase.

## 4.2 Adaptive clamped entropy control

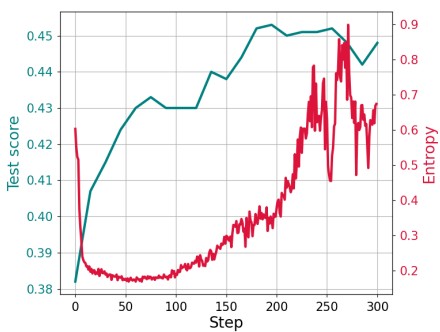

Figure 2: GRPO with a constant entropy bonus coefficient.

For entropy-regularized RL, a constant entropy coefficient $\lambda$ is often sufficient to properly control the policy entropy in robotic and games RL (Mnih et al., 2016; Haarnoja et al., 2018). However, we observe in Figure 2 that this assumption does not necessarily hold in LLM-RL training as the entropy can change drastically in the mid of training, and the initially chosen coefficient fails. In the example, the entropy stabilizes in the early period, but starts to drastically fluctuates after step 200 while the policy performance saturates. The entropy coefficient is not adjusted to change such a trend and fails to deliver better performance promised by entropy control.

To alleviate this issue, we automatically adjust the coefficient during training following

$$\lambda' \leftarrow \text{Proj}_{[\lambda_{\text{low}}, \lambda_{\text{high}}]}\big[\lambda - \beta \min\big(\tilde{\mathcal{H}}(\pi_\theta) - \tilde{\mathcal{H}}_{\text{low}}, 0\big) + \beta \min(\tilde{\mathcal{H}}_{\text{high}} - \tilde{\mathcal{H}}(\pi_\theta), 0)\big] \qquad (4.1)$$

---

**Algorithm 1** AEnt: Adaptive entropy regularization with token space clamping

---

1: Initialize the algorithm, including choosing $\tilde{\mathcal{H}}_{\text{low}}, \tilde{\mathcal{H}}_{\text{high}}$ and $\lambda_{\text{low}}, \lambda_{\text{high}}$, clamping percentage $p$.
2: **for** global step $k = 1$ **to** $K$ **do**
3:     Set the sampling policy $\pi_b$.
4:     Sample a batch of $s_0$ and for each $s_0$, a batch of $(a_0, s_1, a_1, \ldots, s_{H-1}, a_{H-1})$ following $\pi_b$.
5:     Optimize for the batch surrogate of $\mathcal{L}_{\text{AEnt}}(\theta; \lambda)$ w.r.t. $\theta$.
6:     Adjust the clamped entropy coefficient $\lambda$ following scheme 4.1.
7: **end for**

---

where $\beta$ is the coefficient learning rate, and $\tilde{\mathcal{H}}_{\text{low}}, \tilde{\mathcal{H}}_{\text{high}}$ are respectively the lower and upper limit of the (clamped) entropy. The algorithm will try to confine $\tilde{\mathcal{H}}$ within $[\tilde{\mathcal{H}}_{\text{low}}, \tilde{\mathcal{H}}_{\text{high}}]$ by increasing/decreasing $\lambda$ when $\tilde{\mathcal{H}}(\pi_\theta)$ is lower/higher than the limits. The intuition is that when entropy is high, the coefficient should be tuned down to reduce the entropy induced bias and shift weights to reward maximization, which in turn consumes entropy. While when entropy level is too low, the coefficient can be tuned up to leverage the benefits of entropy regularization. For better training stability, the entropy coefficient is also boxed in the range $[\lambda_{\text{low}}, \lambda_{\text{high}}]$ so that large fluctuations of entropy do not lead to coefficient over-shoot. Empirically, we find that this scheme helps improve reasoning efficiency by avoiding entropy and response length explosion.

### 4.3 ALGORITHM

Given the current LLM policy $\pi_\theta$, we approximately maximize the following objective at each step:

$$\mathcal{L}_{\text{AEnt}}(\theta; \lambda) = \mathcal{L}_{\text{PO}}(\theta) + \lambda\tilde{\mathcal{H}}(\pi_\theta) \tag{4.2}$$

where $\mathcal{L}_{\text{PO}}(\theta)$ is a policy optimization objective, e.g., the GRPO objective is used in our tests. At each global step, we set the sampling policy $\pi_b$ according to the choice of policy optimization objective $\mathcal{L}_{\text{PO}}$. For example, in PPO-type algorithms, $\pi_b$ is set as the policy from last global step. Given $\pi_b$, a batch of queries $s_0 \sim \mathcal{D}$ are sampled, and for each query, $\pi_b$ rolls out a batch of trajectories up to the maximum time step. With the batched samples, we can then optimize for the batch surrogate of $\mathcal{L}_{\text{AEnt}}(\theta; \lambda)$ for several mini-epochs. At the end of each global step, the entropy coefficient is adjusted according to scheme 4.1. The whole process is summarized in Algorithm 1.

## 5 EXPERIMENTS

In this section, we conduct experiments to verify the effectiveness of our method.

### 5.1 TRAINING DETAILS

**Models, training datasets and baselines.** The algorithms are tested in multiple training settings: **(a)** we train the Qwen2.5-math-1.5b base model on the MATH dataset (Hendrycks et al., 2021), which contains 7500 math problems with various difficulties and covers multiple mathematical areas; **(b)** we train the DeepSeek-R1-distilled-Qwen-1.5b (DeepSeek-AI, 2025) model on 40k verifiable queries from the OpenR1-math (Open-R1, 2025) dataset, which is derived from Numina-math dataset (Li et al., 2024). In addition, we also train the Qwen2.5-math-7b model on 6k samples from DeepMath dataset (He et al., 2025), the results of which is deferred to Appendix A.6. We compare our algorithm with GRPO and the conventional entropy regularization method which we call EntReg, where the GRPO objective is augmented with the original entropy bonus used in (Mnih et al., 2016; Schulman et al., 2017).

**Evaluation.** We evaluate models on the AIME 2024, MATH-Hard test split (Hendrycks et al., 2021), MATH-500 (Lightman et al., 2023), AMC23, MinervaMath (Lewkowycz et al., 2022) and OlympiadBench (He et al., 2024). We estimate the test score by averaging 4 tries per query on all benchmarks. The test-time generation temperature is 0.6, top-p is 0.95 and top-k is 20.

**Hyper-parameter settings.** The tests are based on the verl framework (Sheng et al., 2025). [1] When training Qwen2.5-math-1.5b base model on the MATH dataset, we use AdamW optimizer with a

---

[1]Our code is available at https://github.com/antgroup/AEnt.

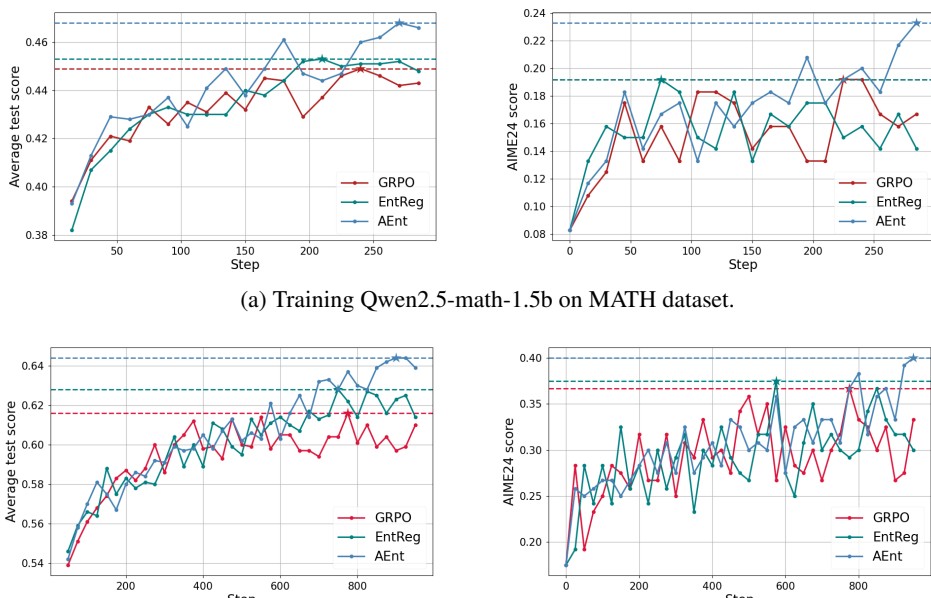

(a) Training Qwen2.5-math-1.5b on MATH dataset.

(b) Training DeepSeek-R1-distilled-Qwen-1.5b on a subset of OpenR1-math dataset.

Figure 3: Test score comparison (see Figure 4 for more training metrics).

Table 1: Test scores by benchmark, where we evaluate the model with the highest average test score trained by each algorithm. Here (a), (b) indicates the two settings described in 5.1. **Bold** numbers indicate the best performance one on the benchmark.

|  | **MATH-Hard** | | **MATH-500** | | **AIME24** | | **Minerva** | | **Olympiad** | | **AMC** | |
|---|---|---|---|---|---|---|---|---|---|---|---|---|
| **Setting** | (a) | (b) | (a) | (b) | (a) | (b) | (a) | (b) | (a) | (b) | (a) | (b) |
| **Base** | 0.368 | 0.661 | 0.584 | 0.792 | 0.083 | 0.225 | 0.179 | 0.311 | 0.279 | 0.432 | 0.406 | 0.594 |
| **GRPO** | 0.524 | 0.773 | 0.756 | 0.865 | 0.192 | 0.367 | 0.311 | 0.347 | 0.364 | 0.576 | 0.550 | 0.769 |
| **EntReg** | 0.546 | 0.808 | **0.752** | 0.872 | 0.167 | 0.342 | 0.316 | **0.359** | 0.370 | 0.576 | 0.562 | 0.794 |
| **AEnt** | **0.552** | **0.813** | 0.750 | **0.882** | **0.217** | **0.392** | **0.330** | **0.359** | **0.377** | **0.591** | **0.581** | **0.825** |

learning rate of $2 \times 10^{-6}$. We set the max response length as $3072$. We use a batch size of $512$, and for each query we roll out $16$ responses with default sampling parameters (top-p and temperature set as 1). For AEnt, we use the GRPO objective as $\mathcal{L}_{\mathrm{PO}}$. We use a clamping percentage $p = 0.33$, and set $\widetilde{\mathcal{H}}_{\mathrm{low}} = 0.15$ and $\widetilde{\mathcal{H}}_{\mathrm{high}} = 0.24$. We use an initial entropy coefficient of $0.002$, and start updating the coefficient from the third epoch with $\beta = 0.002$. We clip the coefficient in between $0.0006$ and $0.009$. For EntReg method, we use the traditional entropy bonus with a fixed entropy coefficient of $0.002$. When training DeepSeek-R1-distilled-Qwen model on the OpenR1-math dataset, we use a learning rate of $1 \times 10^{-6}$, a max response length of $7168$, a batch size of $256$ and for each query we roll out $8$ responses. We use $p = 0.25$, $\widetilde{\mathcal{H}}_{\mathrm{low}} = 0.35$ and $\widetilde{\mathcal{H}}_{\mathrm{high}} = 0.62$, an initial coefficient of $3 \times 10^{-4}$, and start updating the coefficient from the second epoch with $\beta = 10^{-4}$. We clip the coefficient in between $4 \times 10^{-5}$ and $0.001$.

## 5.2 PERFORMANCE ANALYSIS

We report the test performance in Table 1 and Figures 3 & 4. It is observed AEnt outperforms the baselines on average, and on 5 out of the 6 benchmarks across the two different experimental settings.

**An observation on the test score and the entropy trend.** An interesting observation from Figures 4a is that after around 175 steps (collapse time), the policy entropy of GRPO largely depletes and the entropy of EntReg starts to drastically fluctuate, while AEnt's policy entropy is kept stable. Then one

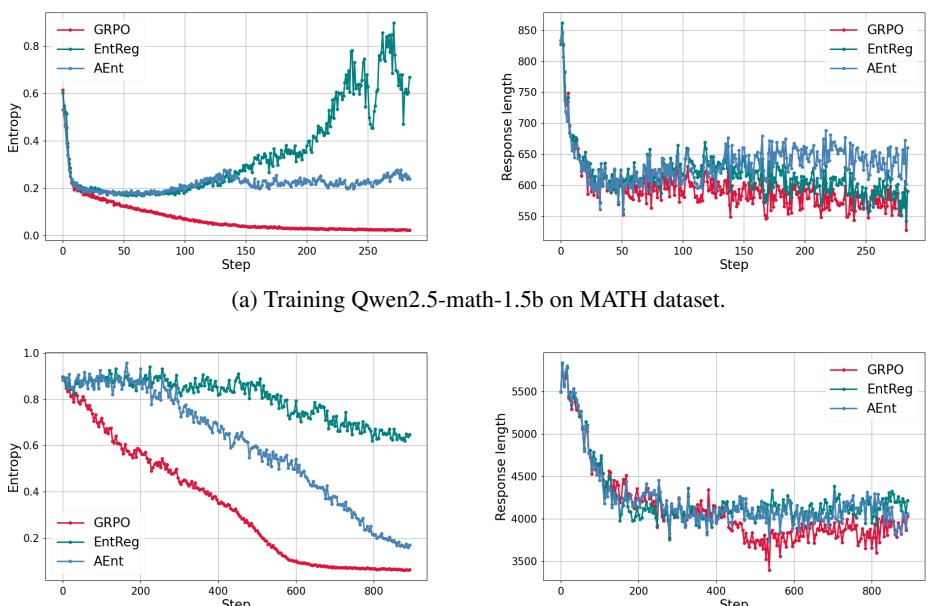

(a) Training Qwen2.5-math-1.5b on MATH dataset.

(b) Training DeepSeek-R1-distilled-Qwen-1.5b on a subset of OpenR1-math dataset.

Figure 4: Entropy and response length trend (see also Figure 3 for test score comparison).

can observe from Figure 3a that the test score of GRPO and EntReg plateaus around the same step, while the score of AEnt continues to improve and surpasses the baselines past the collapse time. This observation is consistent with our intuition and theoretical analysis: after GRPO's entropy collapse, its policy becomes concentrated on few paths and no new information can be gained in the sampling process, ultimately leading to the stagnancy of the learning process. This is predicted by Proposition (I) that the policy will become stationary after entropy depletion. Additionally, it can be observed from Figure 4 that the entropy regularization methods result in slightly longer response in the mid/end of the training period. The potential reason is that the regularization makes the model less certain, and thus the models tend to continue its generation, resulting in longer response. Nonetheless, the increase in response length is relatively mild and we did not observe a major drawback in reasoning efficiency.

## 5.3 ABLATION STUDIES

In this section, we conduct ablation studies on our algorithm.

**Adaptive coefficient stabilizes training.** In Figure 5, we compare the performance of adaptive coefficient vs constant coefficient of the regularizer. The test performance is similar for the two methods in this particular experiment. However, adaptive coefficient leads to a significant advantage on reasoning efficiency by delivering more compact responses while not sacrificing accuracy. In the third plot of Figure 5, constant coefficient fails to stabilize policy entropy in the mid of training, which results in the entropy blow up. We observe a positive correlation between entropy and response length in this case, where a exploding entropy leads to repeated reasoning patterns that do not increase the test scores. On the other hand, the adaptive coefficient successfully prevents the entropy and response length from blowing up.

**Analysis of the entropy clamping percentage $p$.** In Figure 6, we compare the algorithmic performance under different choice of clamping percentage $p$. Intuitively, the percentage $p$ decides the size of the clamped space $\mathcal{A}(s)$, where a larger $p$ leads to more aggressive clamping and less tokens taken into account during entropy calculation. This would smooth the LLM policy on a more compact space, reducing the bias induced by entropy maximization while running the risk to leave out valuable tokens. In this sense, it is reasonable to try to maximize $p$ until the performance drops, which is also suggested by our reported results. Despite the fact the AEnt's performance is affected by the

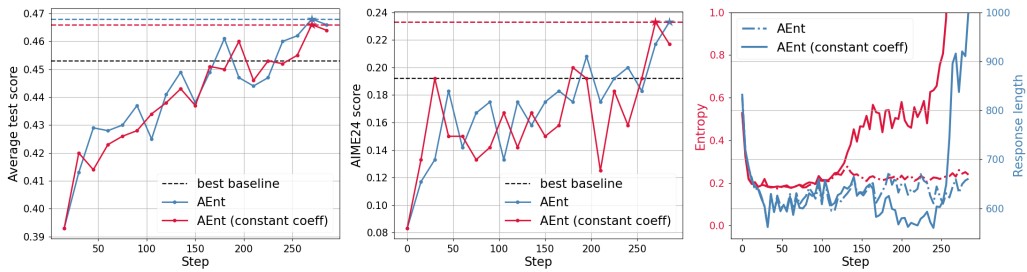

Figure 5: AEnt with adaptive entropy coefficient vs with a constant coefficient. The score in this test is similar. Adaptive coefficient better controls the response length and the policy entropy.

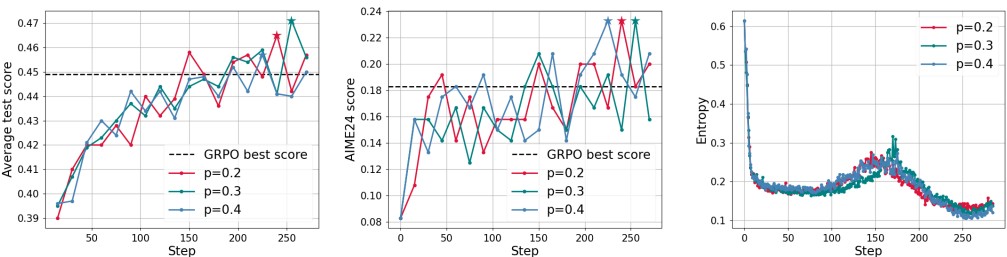

Figure 6: Comparison of different clamping percentage $p$.

choice of $p$, its advantage over the baselines is somewhat robust to the choice. It can be observed from Figure 6 that AEnt outperforms the baselines with different choices of $p$.

## 5.4 TIME COMPLEXITY

We compare the time complexity of each algorithm under different base models and training datasets. The results are reported in Table 2. The experiments on Qwen-2.5-math-1.5b and MATH training dataset are conducted on 4xA100, and all other experiments are conducted on 8xA100. The hyper-paremter setting has been described in Section 5.1 where for fair comparison, we keep the batch sizes, the max response length and all computation speed related configures the same for all algorithms.

Table 2: Time complexity comparison under different settings. "Update per step" indicates the GPU hours of forward/backward process per step; "to GRPO/highest score" indicates the total GPU hours to reach the highest score achieved by GRPO/the algorithm itself. The first and second column respectively reports the results for setting (a) and (b) described in Section 5.1.

|  | Qwen-math-1.5b+MATH | | | R1-distilled-Qwen-1.5b+OpenR1 | | |
|---|---|---|---|---|---|---|
|  | update per step | to GRPO score | to highest score | update per step | to GRPO score | to highest score |
| **GRPO** | 0.234 h | 57 h | 57 h | 0.303 h | 237 h | 237 h |
| **EntReg** | 0.253 h | 49 h | 52 h | 0.316 h | 215 h | 238 h |
| **AEnt** | 0.256 h | 35 h | 69 h | 0.324 h | 186 h | 275 h |

Overall, AEnt consumes slightly more compute per step due to 1) AEnt-trained model's response length is moderately larger than those of the baselines, as indicated in Figure 4; 2) compared to GRPO, it requires the extra forward/backward process of the clamped entropy regularizer, and compared to EntReg, the additional clamping related computation is off-loaded to CPU in our implementation for memory save. However, AEnt reaches a common score threshold faster than the baselines as indicated by the "to GRPO score" results, indicating AEnt preserves the acceleration effect of entropy regularization. AEnt reaches the highest test score slower than the baselines since it prevents premature convergence, and converges to higher test scores which takes more training steps.

## 6 CONCLUSION AND FUTURE DIRECTIONS

In this work, we showed that entropy regularization suffers from large bias in LLM-RL training. As a remedy of this issue, we propose an entropy control method that utilizes a clamped entropy bonus with an automatically adjusted coefficient. We show that AEnt consistently outperforms competitive baselines across multiple benchmarks. We believe AEnt can demonstrate more significant advantages if tested on larger models with more compute. In this work, we did not include a theoretical analysis of the clamped entropy. In addition, we believe the choice of the clamped space $\mathcal{A}(s)$ is crucial to the algorithm's effectiveness, and finding a better choice can potentially yield significant performance gains. For example, one may consider removing actions that are redundant (Baram et al., 2021; Zhong et al., 2024) or grouping similar actions in entropy calculation. Alternatively, since computing entropy on token level suffers from the large dimensionality of LLM vocabulary, it can be beneficial to design entropy regularizer in the state or action representation space (Tennenholtz & Mannor, 2019; Tavakoli et al., 2018). We leave these studies to future works.

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

## A    APPENDIX

### A.1    OMITTED NOTATIONS IN SECTION 2

**Value functions.** We first define some notations we omitted in Section 2. Given the definition of $V^{\pi_\theta})h(s)$, we can define the Q-function as $Q_h^{\pi_\theta}(s,a) = r(s,a) + V_{h+1}^{\pi_\theta}(s')$ with $s' = \mathcal{P}(s,a)$. By this definition, we can also equivalently write

$$Q_h^{\pi_\theta}(s,a) = \mathbb{E}_{\pi_\theta}\Big[\sum_{t=h}^{H} r(s_t, a_t)|s_h = s, a_h = a\Big]$$

We can also define the advantage function as $A_h^{\pi_\theta}(s,a) = Q_h^{\pi_\theta}(s,a) - V_h^{\pi_\theta}(s)$.

**Entropy-regularized value functions.** For the entropy-regularized setting, we can analogously define the entropy-regularized value functions as

$$V_{h,\lambda}^{\pi_\theta}(s) := \mathbb{E}_{\pi_\theta}\Big[\sum_{t=h}^{H-1}\big(r(s_t, a_t) - \lambda\log\pi_\theta(a_t|s_t)\big)|s_h = s\Big]$$

$$Q_{h,\lambda}^{\pi_\theta}(s,a) := r(s,a) + V_{h+1,\lambda}^{\pi_\theta}(s') \text{ with } s' = \mathcal{P}(s,a)$$

Then the entropy-regularized advantage function is defined as $A_{h,\lambda}^{\pi_\theta}(s,a) = Q_{h,\lambda}^{\pi_\theta}(s,a) - \lambda\log\pi_\theta(a|s) - V_{h,\lambda}^{\pi_\theta}(s)$.

Note that we omit the time step subscript in the value functions when $h = 0$, e.g., we write $V_h^{\pi_\theta}|_{h=0}$ as $V^{\pi_\theta}$ and similarly for all the value functions.

## A.2 PRELIMINARY LEMMAS

**Lemma 1** (Entropy gradient). *For the softmax policy, we have*

$$\nabla \mathcal{H}(\pi_\theta) = -\mathbb{E}_{\pi_\theta}\Big[\sum_{h=0}^{H-1} \nabla \log \pi_\theta(a_h|s_h) \sum_{t=h}^{H-1} \log \pi_\theta(a_t|s_t)\Big] \tag{A.1}$$

*Proof.* Starting from the definition of entropy, we can expand the expectation and write

$$\mathcal{H}(\pi_\theta) = -\sum_{s_0,a_0,\dots,a_{H-1}} \mathbb{P}(s_0,a_0\dots,a_{H-1}|\pi_\theta) \sum_{t=0}^{H-1} \log \pi_\theta(a_t|s_t)$$

$$= -\sum_{s_0,a_0,\dots,a_{H-1}} \mathbb{P}(s_0)\pi_\theta(a_0|s_0)\dots\pi_\theta(a_{H-1}|s_{H-1}) \sum_{t=0}^{H-1} \log \pi_\theta(a_t|s_t) \tag{A.2}$$

where in the first equality, the expectation is only taken over $s_0$ and the action sequence since the transition is a deterministic in our LLM setting. Then the gradient of the entropy is given by

$$\nabla \mathcal{H}(\pi_\theta) = -\sum_{s_0,a_0,\dots,a_{H-1}} \mathbb{P}(s_0)\Pi_{h=0}^{H-1}\pi_\theta(a_h|s_h)\nabla\Big(\sum_{t=0}^{H-1} \log \pi_\theta(a_t|s_t)\Big)$$

$$- \sum_{s_0,a_0,\dots,a_{H-1}} \mathbb{P}(s_0)\nabla\Big(\Pi_{h=0}^{H-1}\pi_\theta(a_h|s_h)\Big) \sum_{t=0}^{H-1} \log \pi_\theta(a_t|s_t) \tag{A.3}$$

For the first term in the RHS of equation A.3, we have

$$\sum_{s_0,a_0,\dots,a_{H-1}} \mathbb{P}(s_0)\Pi_{h=0}^{H-1}\pi_\theta(a_h|s_h)\nabla\Big(\sum_{t=0}^{H-1} \log \pi_\theta(a_t|s_t)\Big) = \sum_{s_0,\dots,a_{H-1}} \mathbb{P}(s_0)\nabla\Big(\Pi_{h=0}^{H-1}\pi_\theta(a_h|s_h)\Big)$$

$$= \sum_{s_0} \mathbb{P}(s_0)\nabla\Big(\sum_{a_0\dots a_{H-1}} \Pi_{h=0}^{H-1}\pi_\theta(a_h|s_h)\Big)$$

$$= \sum_{s_0} \mathbb{P}(s_0)\nabla 1 = 0 \tag{A.4}$$

For the second term in the RHS of equation A.3, we have

$$\sum_{s_0,a_0,\dots,a_{H-1}} \mathbb{P}(s_0)\nabla\Big(\Pi_{h=0}^{H-1}\pi_\theta(a_h|s_h)\Big) \sum_{t=0}^{H-1} \log \pi_\theta(a_t|s_t)$$

$$= \sum_{s_0,a_0,\dots,a_{H-1}} \mathbb{P}(s_0)\Pi_{h=0}^{H-1}\pi_\theta(a_h|s_h) \sum_{h=0}^{H-1} \nabla \log \pi_\theta(a_h|s_h) \sum_{t=0}^{H-1} \log \pi_\theta(a_t|s_t)$$

$$= \mathbb{E}_{\pi_\theta}\Big[\sum_{h=0}^{H-1} \nabla \log \pi_\theta(a_h|s_h) \sum_{t=0}^{H-1} \log \pi_\theta(a_t|s_t)\Big]$$

$$= \mathbb{E}_{\pi_\theta}\Big[\sum_{h=1}^{H-1} \nabla \log \pi_\theta(a_h|s_h) \sum_{t=0}^{h-1} \log \pi_\theta(a_t|s_t)\Big] + \mathbb{E}_{\pi_\theta}\Big[\sum_{h=0}^{H-1} \nabla \log \pi_\theta(a_h|s_h) \sum_{t=h}^{H-1} \log \pi_\theta(a_t|s_t)\Big]$$

$$= \mathbb{E}_{\pi_\theta}\Big[\sum_{h=1}^{H-1} \mathbb{E}_{a_h\sim\pi_\theta(s_h)}\big[\nabla \log \pi_\theta(a_h|s_h)|s_h\big] \sum_{t=0}^{h-1} \log \pi_\theta(a_t|s_t)\Big]$$

$$+ \mathbb{E}_{\pi_\theta}\Big[\sum_{h=0}^{H-1} \nabla \log \pi_\theta(a_h|s_h) \sum_{t=h}^{H-1} \log \pi_\theta(a_t|s_t)\Big]$$

$$= \mathbb{E}_{\pi_\theta}\Big[\sum_{h=0}^{H-1} \nabla \log \pi_\theta(a_h|s_h) \sum_{t=h}^{H-1} \log \pi_\theta(a_t|s_t)\Big] \tag{A.5}$$

where the second last equality follows from the towering property of the expectation, and the last equality follows from the fact that for any $s$, we have

$$
\begin{aligned}
\mathbb{E}_{a \sim \pi_\theta(s)}\big[\nabla \log \pi_\theta(a|s)|s\big] &= \sum_a \pi_\theta(a|s)\nabla \log \pi_\theta(a|s) \\
&= \sum_a \nabla \pi_\theta(a|s) \\
&= \nabla \sum_a \pi_\theta(a|s) = \nabla 1 = 0
\end{aligned}
\tag{A.6}
$$

Substituting equation A.4 and equation A.5 into equation A.3 yields

$$
\nabla \mathcal{H}(\pi_\theta) = -\mathbb{E}_{\pi_\theta}\Big[\sum_{h=0}^{H-1} \nabla \log \pi_\theta(a_h|s_h) \sum_{t=h}^{H-1} \log \pi_\theta(a_t|s_t)\Big]
\tag{A.7}
$$

This completes the proof. □

**Lemma 2.** *Given any $h \in \{0, 1, \ldots, H-1\}$ and some baseline functions $b_h^{\pi_\theta} : \mathcal{S} \mapsto \mathbb{R}$, we have for any policy $\pi_\theta$ that:*

$$
\mathbb{E}_{\pi_\theta}\Big[\sum_{h=0}^{H-1} \nabla \log \pi_\theta(a_h|s_h) b_h^{\pi_\theta}(s_h)\Big] = 0
\tag{A.8}
$$

*where the expectation is taken over $(s_0 \sim \mathcal{D}, a_0, \ldots, a_{H-1})$ generated under policy $\pi_\theta$.*

*Proof.* We have

$$
\begin{aligned}
&\mathbb{E}_{\pi_\theta}\Big[\sum_{h=0}^{H-1} \nabla \log \pi_\theta(a_h|s_h) b_h^{\pi_\theta}(s_h)\Big] \\
&= \mathbb{E}_{\pi_\theta}\Big[\sum_{h=0}^{H-1} \mathbb{E}_{a_h \sim \pi_\theta(s_h)}\big[\nabla \log \pi_\theta(a_h|s_h)\big] b_h^{\pi_\theta}(s_h)\Big] = 0
\end{aligned}
\tag{A.9}
$$

which follows from the towering property of the expectation and equation A.6. □

**Lemma 3** (Entropy regularized softmax policy gradient). *If the policy is a softmax, we have*

$$
\nabla_{\theta_{s,a}} V_\lambda^{\pi_\theta}(\mathcal{D}) = \sum_{t=0}^{H-1} \mathbb{P}_t^{\pi_\theta}(s) \pi_\theta(a|s) A_{t,\lambda}^{\pi_\theta}(s, a).
\tag{A.10}
$$

*where $\mathbb{P}_t^{\pi_\theta}(s)$ is the shorthand notation of $\mathbb{P}(s_t = s|\pi_\theta)$, which is the probability of reaching state $s$ at time step $t$ given policy $\pi_\theta$.*

*Proof.* By the policy gradient theorem (Sutton et al., 1999) and its adaptation to the finite-horizon setting (see, e.g., (Klein et al., 2023)), we have

$$
\nabla V^{\pi_\theta}(\mathcal{D}) = \mathbb{E}_{s_0 \sim \mathcal{D}, a_t \sim \pi_\theta(s_t)}\Big[\sum_{t=0}^{H-1} \nabla \log \pi_\theta(a_t|s_t) Q_t^{\pi_\theta}(s_t, a_t)\Big].
\tag{A.11}
$$

The above equality combined with the entropy gradient given in Lemma 1 yields

$$
\begin{aligned}
\nabla V_\lambda^{\pi_\theta}(\mathcal{D}) &= \nabla V_\lambda^{\pi_\theta}(\mathcal{D}) + \lambda \nabla \mathcal{H}(\pi_\theta) \\
&= \mathbb{E}_{\pi_\theta}\Big[\sum_{t=0}^{H-1} \nabla \log \pi_\theta(a_t|s_t)\big(Q_t^{\pi_\theta}(s_t, a_t) - \lambda \sum_{i=t}^{H-1} \log \pi_\theta(a_i|s_i)\big)\Big] \\
&= \mathbb{E}_{\pi_\theta}\Big[\sum_{t=0}^{H-1} \nabla \log \pi_\theta(a_t|s_t)\big(Q_{t,\lambda}^{\pi_\theta}(s_t, a_t) - \lambda \log \pi_\theta(a_t|s_t)\big)\Big].
\end{aligned}
\tag{A.12}
$$

The above equality gives the policy gradient formula with the $Q$-function. It can also be rewritten with the advantage functions. By Lemma 2, we have

$$\mathbb{E}_{\pi_\theta}\Big[\sum_{t=0}^{H-1} V_{t,\lambda}^{\pi_\theta}(s_t)\nabla\log\pi_\theta(a_t|s_t)\Big] = 0. \tag{A.13}$$

Using equation A.13 in equation A.12 gives

$$\nabla V_\lambda^{\pi_\theta}(\mathcal{D}) = \mathbb{E}_{\pi_\theta}\Big[\sum_{t=0}^{H-1}\nabla\log\pi_\theta(a_t|s_t)\big(Q_{t,\lambda}^{\pi_\theta}(s_t,a_t)-\lambda\log\pi_\theta(a_t|s_t)-V_{t,\lambda}^{\pi_\theta}(s_t)\big)\Big]$$

$$= \mathbb{E}_{\pi_\theta}\Big[\sum_{t=0}^{H-1}\nabla\log\pi_\theta(a_t|s_t)A_{t,\lambda}^{\pi_\theta}(s_t,a_t)\Big] \tag{A.14}$$

which follows from the definition of the entropy-regularized advantage function. We can also rewrite the policy gradient formula in equation A.14 with respect to the state marginal distribution as follows:

$$\nabla V_\lambda^{\pi_\theta}(\mathcal{D}) = \mathbb{E}_{s_0\sim\mathcal{D},a_t\sim\pi_\theta(s_t)}\Big[\sum_{t=0}^{H-1}\nabla\log\pi_\theta(a_t|s_t)A_{t,\lambda}^{\pi_\theta}(s_t,a_t)\Big]$$

$$= \sum_{t=0}^{H-1}\mathbb{E}_{s\sim\mathbb{P}_t^{\pi_\theta},a\sim\pi_\theta(s)}\Big[\nabla\log\pi_\theta(a|s)A_{t,\lambda}^{\pi_\theta}(s,a)\Big]$$

$$= \sum_{t=0}^{H-1}\sum_s\mathbb{P}_t^{\pi_\theta}(s)\sum_a\pi_\theta(a|s)\nabla\log\pi_\theta(a|s)A_{t,\lambda}^{\pi_\theta}(s,a) \tag{A.15}$$

Under the softmax policy, we have $\nabla_{\theta_{\bar{s},\bar{a}}}\log\pi_\theta(a|s) = \mathbf{1}_{s=\bar{s}}\big(\mathbf{1}_{a=\bar{a}}-\pi_\theta(\bar{a}|\bar{s})\big)$. Then the element-wise policy gradient is

$$\nabla_{\theta_{\bar{s},\bar{a}}}V_\lambda^{\pi_\theta}(\mathcal{D}) = \sum_{t=0}^{H-1}\sum_s\mathbb{P}_t^{\pi_\theta}(s)\sum_a\pi_\theta(a|s)\mathbf{1}_{s=\bar{s}}\Big(\mathbf{1}_{a=\bar{a}}-\pi_\theta(\bar{a}|\bar{s})\Big)A_{t,\lambda}^{\pi_\theta}(s,a)$$

$$= \sum_{t=0}^{H-1}\mathbb{P}_t^{\pi_\theta}(\bar{s})\sum_a\pi_\theta(a|\bar{s})\Big(\mathbf{1}_{a=\bar{a}}-\pi_\theta(\bar{a}|\bar{s})\Big)A_{t,\lambda}^{\pi_\theta}(\bar{s},a)$$

$$= \sum_{t=0}^{H-1}\mathbb{P}_t^{\pi_\theta}(\bar{s})\pi_\theta(\bar{a}|\bar{s})A_{t,\lambda}^{\pi_\theta}(\bar{s},\bar{a}). \tag{A.16}$$

where the last inequality is due to $\mathbb{E}_{a\sim\pi_\theta(s)}[A_{t,\lambda}^{\pi_\theta}(s,a)] = 0$ following the definition of the value functions. $\qquad\square$

**Lemma 4** (Performance difference lemma). *We have for any $h\in\{0,1,\ldots,H-1\}$ and state $s\in\mathcal{S}$, the performance difference between any two policies $\pi$ and $\pi'$ is*

$$V_h^\pi(s) - V_h^{\pi'}(s) = \mathbb{E}_\pi\Big[\sum_{t=h}^{H-1} A_t^{\pi'}(s_t,a_t)|s_h = s\Big]. \tag{A.17}$$

*Proof.* We have

$$V_h^\pi(s) - V_h^{\pi'}(s)$$

$$= \mathbb{E}_\pi \Big[ \sum_{t=h}^{H-1} r(s_t, a_t) | s_h = s \Big] - V_h^{\pi'}(s)$$

$$= \mathbb{E}_\pi \Big[ \sum_{t=h}^{H-1} r(s_t, a_t) + \sum_{t=h}^{H-2} V_{t,\lambda}^{\pi'}(s_{t+1}) - \sum_{t=h}^{H-2} V_{t,\lambda}^{\pi'}(s_{t+1}) | s_h = s \Big] - V_h^{\pi'}(s)$$

$$= \mathbb{E}_\pi \Big[ \sum_{t=h}^{H-1} Q_{t,\lambda}^{\pi'}(s_t, a_t) - \sum_{t=h}^{H-1} V_{t,\lambda}^{\pi'}(s_t) | s_h = s \Big]$$

$$= \mathbb{E}_\pi \Big[ \sum_{t=h}^{H-1} A_t^{\pi'}(s_t, a_t) | s_h = s \Big] \tag{A.18}$$

This completes the proof. $\qquad\square$

## A.3 PROOF OMITTED IN SECTION 3

### A.3.1 PROOF OF PROPOSITION 1

*Proof.* We start with proving the first bullet. Denote the entropy of $\pi_\theta(\cdot|s)$ as

$$\mathcal{H}(\pi(\cdot|s)) = - \sum_a \pi_\theta(a|s) \log \pi_\theta(a|s) \tag{A.19}$$

Since $1 - x \leq -\log x$ for $0 < x \leq 1$, we have

$$\mathcal{H}(\pi(\cdot|s)) \geq \sum_a \pi_\theta(a|s)(1 - \pi_\theta(a|s)) \tag{A.20}$$

Viewing $\pi_\theta(\cdot|s)$ as a vector in $\Delta^{|\mathcal{A}|}$, it is known that the softmax Jacobian can be written as

$$\frac{\partial \pi_\theta(\cdot|s)}{\partial \theta_{s,\cdot}} = \text{Diag}(\pi_\theta(\cdot|s)) - \pi_\theta(\cdot|s)\pi_\theta(\cdot|s)^\top \tag{A.21}$$

Then we have

$$\Big\| \frac{\partial \pi_\theta(\cdot|s)}{\partial \theta_{s,\cdot}} \Big\| \leq \Big\| \frac{\partial \pi_\theta(\cdot|s)}{\partial \theta_{s,\cdot}} \Big\|_F$$

$$\leq \sum_a \Big( \pi_\theta(a|s)\big(1 - \pi_\theta(a|s)\big) + \pi_\theta(a|s) \sum_{a'} \pi_\theta(a'|s) \Big)$$

$$= 2 \sum_a \pi_\theta(a|s)\big(1 - \pi_\theta(a|s)\big)$$

$$\leq 2\mathcal{H}(\pi_\theta(\cdot|s)) \tag{A.22}$$

By equation A.15 in Lemma 3, we have

$$\nabla V_\lambda^{\pi_\theta}(\mathcal{D}) = \sum_{t=0}^{H-1} \sum_s \mathbb{P}_t^{\pi_\theta}(s) \sum_a \pi_\theta(a|s) \nabla \log \pi_\theta(a|s) A_{t,\lambda}^{\pi_\theta}(s, a)$$

$$= \sum_{t=0}^{H-1} \sum_s \mathbb{P}_t^{\pi_\theta}(s) \sum_a \nabla \pi_\theta(a|s) A_{t,\lambda}^{\pi_\theta}(s, a)$$

$$= \sum_{t=0}^{H-1} \sum_s \mathbb{P}_t^{\pi_\theta}(s) \sum_a \frac{\partial \pi_\theta(a|s)}{\partial \theta_{s,\cdot}} A_{t,\lambda}^{\pi_\theta}(s, a)$$

$$= \sum_{t=0}^{H-1} \sum_s \mathbb{P}_t^{\pi_\theta}(s) \frac{\partial \pi_\theta(\cdot|s)}{\partial \theta_{s,\cdot}} A_{t,\lambda}^{\pi_\theta}(s, \cdot) \tag{A.23}$$

where the third equality is due to $\frac{\partial \pi_\theta(a|s)}{\partial \theta_{s',\cdot}} = 0$ if $s' \neq s$. Then we have

$$
\begin{aligned}
\|\nabla V_\lambda^{\pi_\theta}(\mathcal{D})\| &\leq \sum_{t=0}^{H-1} \sum_s \mathbb{P}_t^{\pi_\theta}(s) \left\| \frac{\partial \pi_\theta(\cdot|s)}{\partial \theta_{s,\cdot}} \right\| \\
&\leq 2 \sum_{t=0}^{H-1} \sum_s \mathbb{P}_t^{\pi_\theta}(s) \mathcal{H}(\pi_\theta(\cdot|s)) \\
&= 2\mathcal{H}(\pi_\theta)
\end{aligned}
\tag{A.24}
$$

where the last inequality is due to the definition of the policy entropy:

$$
\begin{aligned}
\mathcal{H}(\pi_\theta) &= -\mathbb{E}_{\pi_\theta} \Big[ \sum_{t=0}^{H-1} \log \pi_\theta(a_t|s_t) \big| s_0 \sim \mathcal{D} \Big] \\
&= - \sum_{t=0}^{H-1} \sum_s \mathbb{P}_t^{\pi_\theta}(s) \sum_a \pi_\theta(a|s) \log \pi_\theta(a|s)
\end{aligned}
\tag{A.25}
$$

This completes the proof of the first bullet.

Next we provide the proof of the second bullet. Let $\pi^* \in \arg\max_\pi V^\pi(\mathcal{D})$ be any deterministic optimal policy. Given any $s_0 \sim \mathcal{D}$, let $s_h^*, a_h^*$ be a state-action pair generated by $\pi^*$ up to time step $h$, e.g., $a_h^* = \pi^*(s_h^*)$. We write $s_0^* = s_0$.

Given any $s_0$, we have

$$
\begin{aligned}
\|\nabla V^{\pi_\theta}(\mathcal{D})\| &\geq \Big( \sum_{h=0}^{H-1} \big( \nabla_{s_h^*, a_h^*} V^{\pi_\theta}(\mathcal{D}) \big)^2 \Big)^{0.5} \\
&\geq \frac{1}{\sqrt{H}} \sum_{h=0}^{H-1} \big| \nabla_{s_h^*, a_h^*} V^{\pi_\theta}(\mathcal{D}) \big| \\
&= \frac{1}{\sqrt{H}} \sum_{h=0}^{H-1} \sum_{t=0}^{H-1} \mathbb{P}_t^{\pi_\theta}(s_h^*) \pi_\theta(a_h^*|s_h^*) \big| A_t^{\pi_\theta}(s_h^*, a_h^*) \big|
\end{aligned}
\tag{A.26}
$$

where the second inequality follows from Cauchy-Schwartz inequality, and the equality follows from the softmax policy gradient derived in Lemma 3.

Given $s_0$, by the assumption of our LLM tasks that $s_{t+1} = \mathcal{P}(s_t, a_t)$ is a concatenation of $s_t, a_t$ for any $0 \leq t \leq H-1$, we have $\mathbb{P}_t^{\pi_\theta}(s_h^*) = 0$ for any $t \neq h$. Using this fact in equation A.26

$$
\|\nabla V^{\pi_\theta}(\mathcal{D})\| \geq \frac{1}{\sqrt{H}} \sum_{h=0}^{H-1} \mathbb{P}_h^{\pi_\theta}(s_h^*) \pi_\theta(a_h^*|s_h^*) | A_h^{\pi_\theta}(s_h^*, a_h^*) |
\tag{A.27}
$$

Continuing from equation A.27,

$$
\begin{aligned}
&\|\nabla V^{\pi_\theta}(\mathcal{D})\| \\
&\geq \frac{1}{\sqrt{H}} \sum_{h=0}^{H-1} \mathbb{P}_h^{\pi_\theta}(s_h^*) \pi_\theta(a_h^*|s_h^*) | A_h^{\pi_\theta}(s_h^*, a_h^*) | \\
&= \frac{1}{\sqrt{H}} \sum_{h=0}^{H-1} \frac{\mathbb{P}_h^{\pi_\theta}(s_h^*) \pi_\theta(a_h^*|s_h^*)}{\mathbb{P}_h^{\pi^*}(s_h^*) \pi^*(a_h^*|s_h^*)} \mathbb{P}_h^{\pi^*}(s_h^*) \pi^*(a_h^*|s_h^*) | A_h^{\pi_\theta}(s_h^*, a_h^*) | \\
&= \frac{1}{\sqrt{H}} \sum_{h=0}^{H-1} \frac{\mathbb{P}_h^{\pi_\theta}(s_h^*) \pi_\theta(a_h^*|s_h^*)}{\mathbb{P}(s_0) \pi^*(a_0^*|s_0^*) \pi^*(a_1^*|s_1^*) \ldots \pi^*(a_h^*|s_h^*)} \mathbb{P}_h^{\pi^*}(s_h^*) \pi^*(a_h^*|s_h^*) | A_h^{\pi_\theta}(s_h^*, a_h^*) | \\
&= \frac{|\mathcal{D}|}{\sqrt{H}} \sum_{h=0}^{H-1} \mathbb{P}_h^{\pi_\theta}(s_h^*) \pi_\theta(a_h^*|s_h^*) \mathbb{P}_h^{\pi^*}(s_h^*) \pi^*(a_h^*|s_h^*) | A_h^{\pi_\theta}(s_h^*, a_h^*) |
\end{aligned}
\tag{A.28}
$$

where the second last inequality follows from the definition of $\mathbb{P}_h^\pi(s_h)$, and the last inequality follows from the fact that $\pi^*$ is defined as a deterministic optimal policy, yielding

$$\mathbb{P}(s_0)\pi^*(a_0^*|s_0^*)\pi^*(a_1^*|s_1^*)\ldots\pi^*(a_h^*|s_h^*) = \mathbb{P}(s_0) = \frac{1}{|\mathcal{D}|}. \tag{A.29}$$

Continuing from equation A.28, we have

$$\|\nabla V^{\pi_\theta}(\mathcal{D})\|$$

$$\geq \Big(\min_{h\in\{0,1,\ldots,H-1\}}\mathbb{P}_h^{\pi_\theta}(s_h^*)\pi_\theta(a_h^*|s_h^*)\Big)\frac{|\mathcal{D}|}{\sqrt{H}}\sum_{h=0}^{H-1}\mathbb{P}_h^{\pi^*}(s_h^*)\pi^*(a_h^*|s_h^*)A_h^{\pi_\theta}(s_h^*,a_h^*)$$

$$= \Pi_{h=0}^{H-1}\pi_\theta(a_h^*|s_h^*)\frac{1}{\sqrt{H}}\sum_{h=0}^{H-1}\mathbb{P}_h^{\pi^*}(s_h^*)\pi^*(a_h^*|s_h^*)A_h^{\pi_\theta}(s_h^*,a_h^*)$$

$$\geq \frac{1}{\sqrt{H}|\mathcal{D}|}\Pi_{h=0}^{H-1}\pi_\theta(a_h^*|s_h^*)\mathbb{E}_{\pi^*}\big[A_h^{\pi_\theta}(s_h^*,a_h^*)|s_0\big]$$

$$\geq \frac{1}{\sqrt{H}|\mathcal{D}|}\Pi_{h=0}^{H-1}\pi_\theta(a_h^*|s_h^*)\Big(V^{\pi^*}(s_0) - V^{\pi_\theta}(s_0)\Big). \tag{A.30}$$

Note that this inequality holds for any trajectory $(s_0, a_0^*, a_1^*, \ldots, a_{H-1}^*)$ generated by any deterministic optimal policy $\pi^*$. Then we have

$$\|\nabla V^{\pi_\theta}(\mathcal{D})\| \geq C^{\pi_\theta}(s_0)\Big(V^{\pi^*}(s_0) - V^{\pi_\theta}(s_0)\Big) \tag{A.31}$$

where $C^{\pi_\theta}(s_0) = \frac{1}{\sqrt{H}|\mathcal{D}|}\max_{(a_0,\ldots,a_{H-1})\in\mathcal{A}_H^*(s_0)}\Pi_{t=0}^{H-1}\pi_\theta(a_t|s_t)$ with $\mathcal{A}_H^*(s_0) = \{(a_0, a_1, \ldots, a_{H-1}) \in \mathcal{A}^H \mid \exists \pi^* \in \arg\max_\pi V^\pi(\mathcal{D}), \Pi_{t=0}^{H-1}\pi^*(a_t|s_t) > 0\}$. $\qquad\square$

## A.4 PROOF OF PROPOSITION 2

Proposition 2 can be proven by combining Lemma 5 and Lemma 6.

**Lemma 5.** *It holds that*

$$V^{\pi^*}(s_0) - V^{\pi_\theta}(s_0) \leq V_\lambda^{\pi_\lambda^*}(s_0) - V_\lambda^{\pi_\theta}(s_0) + \lambda H \log\frac{|\mathcal{A}|}{|\mathcal{A}_H^*(s_0)|^{\frac{1}{H}}} \tag{A.32}$$

*where $\pi_\lambda^* = \arg\max_\pi V_\lambda^\pi(\mathcal{D})$, and recall $\pi^* \in \arg\max_\pi V^\pi(\mathcal{D})$. Here $\mathcal{A}_H^*(s_0) = \{(a_0, a_1, \ldots, a_{H-1}) \in \mathcal{A}^H \mid \exists \pi^*, \Pi_{t=0}^{H-1}\pi^*(a_t|s_t) > 0\}$ is the set of all optimal responses given query $s_0$.*

*Proof.* Define $\mathcal{H}(\pi|s_0)$ as

$$\mathcal{H}(\pi|s_0) = -\mathbb{E}_\pi\Big[\sum_{t=0}^{H-1}\log\pi(a_t|s_t)|s_0\Big] \tag{A.33}$$

For any $\pi^* \in \arg\max_\pi V^\pi(\mathcal{D})$, by the optimality of $\pi_\lambda^*$ we have

$$V_\lambda^{\pi_\lambda^*}(s_0) - V_\lambda^{\pi_\theta}(s_0) \geq V_\lambda^{\pi^*}(s_0) - V_\lambda^{\pi_\theta}(s_0)$$
$$= V^{\pi^*}(s_0) - V^{\pi_\theta}(s_0) + \lambda(\mathcal{H}(\pi^*|s_0) - \mathcal{H}(\pi_\theta|s_0)) \tag{A.34}$$

where the equality follows from the definition of $V_\lambda^\pi(s_0)$. Then we have

$$V_\lambda^{\pi_\lambda^*}(s_0) - V_\lambda^{\pi_\theta}(s_0)$$
$$\geq V^{\pi^*}(s_0) - V^{\pi_\theta}(s_0) + \lambda\Big(\max_{\pi^*\in\arg\max_\pi V^\pi(\mathcal{D})}\mathcal{H}(\pi^*|s_0) - \mathcal{H}(\pi_\theta|s_0)\Big) \tag{A.35}$$

Notice that

$$
\begin{aligned}
\max_{\pi^*} \mathcal{H}(\pi^*|s_0) &= \max_{\pi^*} -\mathbb{E}_{\pi^*}\Big[ \sum_{t=0}^{H-1} \log \pi^*(a_t|s_t)|s_0 \Big] \\
&= \max_{\pi^*} - \sum_{a_0,\dots,a_{H-1}\in\mathcal{A}_H^*(s_0)} \Pi_{t=0}^{H-1}\pi^*(a_t|s_t)\Big[ \log \Pi_{t=0}^{H-1}\pi^*(a_t|s_t) \Big] \\
&\leq \max_{\mathbb{P}\in\Delta(\mathcal{A}_H^*(s_0))} - \sum_{\tau\in\mathcal{A}_H^*(s_0)} \mathbb{P}(\tau)\Big[ \log \mathbb{P}(\tau) \Big] \\
&= - \sum_{\tau\in\mathcal{A}_H^*(s_0)} \frac{1}{|\mathcal{A}_H^*(s_0)|} \log \frac{1}{|\mathcal{A}_H^*(s_0)|} \\
&= \log |\mathcal{A}_H^*(s_0)|. \qquad\qquad\qquad\qquad\qquad\qquad\qquad\qquad (A.36)
\end{aligned}
$$

where in the third inequality, $\Delta(\mathcal{A}_H^*(s_0))$ denotes the probability simplex on $\mathcal{A}_H^*(s_0)$. Additionally, it is known that

$$
\mathcal{H}(\pi_\theta|s_0) \leq \max_\pi \mathcal{H}(\pi|s_0) = H \log |\mathcal{A}| \qquad\qquad (A.37)
$$

Substituting equation A.36 and equation A.37 into equation A.35 yields

$$
V_\lambda^{\pi_\lambda^*}(s_0) - V_\lambda^{\pi_\theta}(s_0) \geq V^{\pi^*}(s_0) - V^{\pi_\theta}(s_0) + \lambda H \log \frac{|\mathcal{A}|}{|\mathcal{A}_H^*(s_0)|^{\frac{1}{H}}} \qquad (A.38)
$$

which completes the proof. $\qquad\qquad\qquad\qquad\qquad\qquad\qquad\qquad\qquad\qquad\qquad\qquad\quad\square$

Next we present the performance bound under entropy regularization. The derivation is adapted from (Mei et al., 2020, Lemma 15) for the LLM setting modeled as finite-horizon MDPs with a deterministic state transition.

**Lemma 6.** *Assume the policy is a softmax. Then it holds that*

$$
V_\lambda^{\pi_\lambda^*}(s_0) - V_\lambda^{\pi_\theta}(s_0) \leq \frac{1}{2\lambda} \frac{1}{C_\lambda^{\pi_\theta}(s_0)} \|\nabla V_\lambda^{\pi_\theta}(\mathcal{D})\|^2 \qquad (A.39)
$$

*where and $C_\lambda^{\pi_\theta}(s_0)$ is specified in the proof.*

*Proof.* The performance gap can be bounded as

$$
\begin{aligned}
&V_\lambda^{\pi_\lambda^*}(s_0) - V_\lambda^{\pi_\theta}(s_0) \\
&= \mathbb{E}_{\pi_\lambda^*}\Big[ \sum_{t=0}^{H-1} r(s_t,a_t) - \lambda \log \pi_\lambda^*(a_t|s_t) + V_{t,\lambda}^{\pi_\theta}(s_t) - V_{t,\lambda}^{\pi_\theta}(s_t)|s_0 \Big] - V_{t,\lambda}^{\pi_\theta}(s_0) \\
&= \mathbb{E}_{\pi_\lambda^*}\Big[ \sum_{t=0}^{H-1} r(s_t,a_t) - \lambda \log \pi_\lambda^*(a_t|s_t) + V_{t+1,\lambda}^{\pi_\theta}(s_{t+1}) - V_{t,\lambda}^{\pi_\theta}(s_t)|s_0 \Big] \\
&= \mathbb{E}_{\pi_\lambda^*}\Big[ \sum_{t=0}^{H-1} Q_{t,\lambda}^{\pi_\theta}(s_t,a_t) - \lambda \log \pi_\lambda^*(a_t|s_t) - V_{t,\lambda}^{\pi_\theta}(s_t)|s_0 \Big] \\
&= \sum_{t=0}^{H-1} \mathbb{E}_{s\sim\mathbb{P}_t^{\pi_\lambda^*}(\cdot|s_0)}\Big[ \mathbb{E}_{a\sim\pi_\lambda^*(s)}\big[ Q_{t,\lambda}^{\pi_\theta}(s,a) - \lambda \log \pi_\lambda^*(a|s) \big] - V_{t,\lambda}^{\pi_\theta}(s) \Big] \qquad (A.40)
\end{aligned}
$$

where $\mathbb{P}_t^{\pi_\lambda^*}(\cdot|s_0) = \mathbb{P}(s_t = \cdot|s_0, \pi_\lambda^*)$ is the probability distribution of $s_t$ under policy $\pi_\lambda^*$ given the initial state $s_0$. The second last equality uses the definition of $Q_{t,\lambda}^{\pi_\theta}$ that $Q_{t,\lambda}^{\pi_\theta}(s_t,a_t) = r(s_t,a_t) + V_{t+1,\lambda}^{\pi_\theta}(s_{t+1})$ with $s_{t+1} = (s_t,a_t)$.

Given any $s$, we have

$$
\mathbb{E}_{a \sim \pi_\lambda^*(s)} \Big[ Q_{t,\lambda}^{\pi_\theta}(s,a) - \lambda \log \pi_\lambda^*(a|s) \Big] \leq \max_\pi \sum_a \pi(a|s) Q_{t,\lambda}^{\pi_\theta}(s,a) - \lambda \pi(a|s) \log \pi(a|s)
$$

$$
= \sum_a \bar{\pi}_\theta(a|s,t) Q_{t,\lambda}^{\pi_\theta}(s,a) - \lambda \bar{\pi}_\theta(a|s,t) \log \bar{\pi}_\theta(a|s,t)
$$

$$
= \lambda \log \sum_a \exp(Q_{t,\lambda}^{\pi_\theta}(s,a)/\lambda) \tag{A.41}
$$

where $\bar{\pi}_\theta(a|s,t) = \exp\left(Q_{t,\lambda}^{\pi_\theta}(s,a)/\lambda\right) / \sum_a \exp\left(Q_{t,\lambda}^{\pi_\theta}(s,a)/\lambda\right)$. Notice that

$$
V_{t,\lambda}^{\pi_\theta}(s) = \sum_a \pi_\theta(a|s) \big( Q_{t,\lambda}^{\pi_\theta}(s,a) - \lambda \log \pi_\theta(a|s) \big)
$$

$$
= \sum_a \pi_\theta(a|s) \big( Q_{t,\lambda}^{\pi_\theta}(s,a) - \lambda \log \pi_\theta(a|s) + \lambda \log \bar{\pi}_\theta(a|s,t) - \lambda \log \bar{\pi}_\theta(a|s,t) \big)
$$

$$
= \lambda \log \sum_a \exp(Q_{t,\lambda}^{\pi_\theta}(s,a)/\lambda) - \lambda D_{\mathrm{KL}}(\pi_\theta(s,t) || \bar{\pi}_\theta(s,t)) \tag{A.42}
$$

Substituting equation A.41 and equation A.42 into equation A.40 yields

$$
V_\lambda^{\pi_\lambda^*}(s_0) - V_{t,\lambda}^{\pi_\theta}(s_0)
$$

$$
\leq \sum_{t=0}^{H-1} \mathbb{E}_{s \sim \mathbb{P}_t^{\pi_\lambda^*}(\cdot|s_0)} \Big[ D_{\mathrm{KL}}(\pi_\theta(s,t) || \bar{\pi}_\theta(s,t)) \Big]
$$

$$
\leq \frac{\lambda}{2} \sum_{t=0}^{H-1} \mathbb{E}_{s \sim \mathbb{P}_t^{\pi_\lambda^*}(\cdot|s_0)} \left\| \frac{Q_{t,\lambda}^{\pi_\theta}(s,\cdot)}{\lambda} - \theta_{s,\cdot} - \frac{\sum_a Q_{t,\lambda}^{\pi_\theta}(s,a)/\lambda - \theta_{s,a}}{|\mathcal{A}|} \mathbf{1} \right\|_\infty^2 \tag{A.43}
$$

where $\mathbf{1} \in \mathbb{R}^{|\mathcal{A}|}$ is an all-one vector and the last inequality follows from (Mei et al., 2020, Lemma 27).

Following the derivation of Lemma 3, it is straightforward to verify that equation A.15 holds with $Q_{t,\lambda}^{\pi_\theta}(s,a) - \lambda \log \pi_\theta(a|s)$ in place of the advantage $A_{t,\lambda}^{\pi_\theta}$:

$$
\nabla V_\lambda^{\pi_\theta}(\mathcal{D}) = \sum_{t=0}^{H-1} \sum_s \mathbb{P}_t^{\pi_\theta}(s) \sum_a \pi_\theta(a|s) \nabla \log \pi_\theta(a|s) \big( Q_{t,\lambda}^{\pi_\theta}(s,a) - \lambda \log \pi_\theta(a|s) \big)
$$

$$
= \sum_{t=0}^{H-1} \sum_s \mathbb{P}_t^{\pi_\theta}(s) \sum_a \nabla \pi_\theta(a|s) \big( Q_{t,\lambda}^{\pi_\theta}(s,a) - \lambda \log \pi_\theta(a|s) \big)
$$

$$
= \sum_{t=0}^{H-1} \sum_s \mathbb{P}_t^{\pi_\theta}(s) \sum_a \nabla \pi_\theta(a|s) \big( Q_{t,\lambda}^{\pi_\theta}(s,a) - \lambda \theta_{s,a} + \lambda \sum_a \exp \theta_{s,a} \big)
$$

$$
= \sum_{t=0}^{H-1} \sum_s \mathbb{P}_t^{\pi_\theta}(s) \sum_a \nabla \pi_\theta(a|s) \big( Q_{t,\lambda}^{\pi_\theta}(s,a) - \lambda \theta_{s,a} \big) \tag{A.44}
$$

where third equality follows from the $\pi_\theta(a|s)$ is a softmax function, and the last equality is due to the fact that

$$
\sum_a \nabla \pi_\theta(a|s) \sum_a \exp \theta_{s,a} = \sum_a \exp \theta_{s,a} \nabla \sum_a \pi_\theta(a|s) = \sum_a \exp \theta_{s,a} \nabla 1 = 0.
$$

Then from equation A.44, we have

$$
\frac{\partial V_\lambda^{\pi_\theta}(\mathcal{D})}{\partial \theta_{s,\cdot}} = \sum_{t=0}^{H-1} \mathbb{P}_t^{\pi_\theta}(s) \sum_a \frac{\partial \pi_\theta(a|s)}{\partial \theta_{s,\cdot}} \big( Q_{t,\lambda}^{\pi_\theta}(s,a) - \lambda \theta_{s,a} \big)
$$

$$
= \sum_{t=0}^{H-1} \mathbb{P}_t^{\pi_\theta}(s) \frac{\partial \pi_\theta(\cdot|s)}{\partial \theta_{s,\cdot}} \big( Q_{t,\lambda}^{\pi_\theta}(s,\cdot) - \lambda \theta_{s,\cdot} \big). \tag{A.45}
$$

where the first equality is due to the fact that $\partial \pi_\theta(a|s')/\partial \theta_{s,\cdot} = 0$ for any $s' \neq s$, and the last equality follows from a matrix-vector product rewriting.

Define $\mathcal{S}(s_0) \subseteq \mathcal{S}$ as the set of all possible states starting from $s_0$, i.e., $\mathcal{S}(s_0) = \{s_0\} \cup \{s_t \in \mathcal{S} | t \in \{1, ..., H-1\}, a_{t-1} \in \mathcal{A}, s_{t-1} \in \mathcal{S}(s_0), s_t = \mathcal{P}(s_{t-1}, a_{t-1})\}$. Then we have

$$
\begin{aligned}
\|\nabla V_\lambda^{\pi_\theta}(\mathcal{D})\| &\geq \Big( \sum_{s \in \mathcal{S}(s_0)} \|\frac{\partial V_\lambda^{\pi_\theta}(\mathcal{D})}{\partial \theta_{s,\cdot}}\|^2 \Big)^{0.5} \\
&\geq \frac{1}{\sqrt{|\mathcal{S}(s_0)|}} \sum_{s \in \mathcal{S}(s_0)} \|\frac{\partial V_\lambda^{\pi_\theta}(\mathcal{D})}{\partial \theta_{s,\cdot}}\| \\
&= C_d \sum_{s \in \mathcal{S}(s_0)} \sum_{t=0}^{H-1} \mathbb{P}_t^{\pi_\theta}(s) \|\frac{\partial \pi_\theta(\cdot|s)}{\partial \theta_{s,\cdot}} \big( Q_{t,\lambda}^{\pi_\theta}(s,\cdot) - \lambda \theta_{s,\cdot} \big) \|
\end{aligned}
\tag{A.46}
$$

where the second and the third inequalities follow from Cauchy-Schwartz inequality, and the last inequality follows from equation A.45. The constant $C_d = \frac{1}{\sqrt{|\mathcal{S}(s_0)|}}$.

Continuing from equation A.46, it follows similar to the derivations in (533)–(536) in (Mei et al., 2020) that

$$
\begin{aligned}
&\|\nabla V_\lambda^{\pi_\theta}(\mathcal{D})\| \\
&\geq C_d \sum_{s \in \mathcal{S}(s_0)} \sum_{t=0}^{H-1} \mathbb{P}_t^{\pi_\theta}(s) \min_a \pi_\theta(a|s) \Big\| Q_{t,\lambda}^{\pi_\theta}(s,\cdot) - \lambda \theta_{s,\cdot} - \frac{\sum_a Q_{t,\lambda}^{\pi_\theta}(s,a) - \lambda \theta_{s,a}}{|\mathcal{A}|} \Big\|_\infty
\end{aligned}
\tag{A.47}
$$

Then we have

$$
\begin{aligned}
&\|\nabla V_\lambda^{\pi_\theta}(\mathcal{D})\|^2 \\
&\geq C_d^2 \sum_{s \in \mathcal{S}(s_0)} \sum_{t=0}^{H-1} (\mathbb{P}_t^{\pi_\theta}(s) \min_a \pi_\theta(a|s))^2 \Big\| Q_{t,\lambda}^{\pi_\theta}(s,\cdot) - \lambda \theta_{s,\cdot} - \frac{\sum_a Q_{t,\lambda}^{\pi_\theta}(s,a) - \lambda \theta_{s,a}}{|\mathcal{A}|} \Big\|_\infty^2 \\
&= C_d^2 \lambda^2 \sum_{s \in \mathcal{S}(s_0)} \sum_{t=0}^{H-1} \frac{(\mathbb{P}_t^{\pi_\theta}(s) \min_a \pi_\theta(a|s))^2}{\mathbb{P}_t^{\pi_\lambda^*}(s|s_0)} \mathbb{P}_t^{\pi_\lambda^*}(s|s_0) \Big\| Q_{t,\lambda}^{\pi_\theta}(s,\cdot)/\lambda - \theta_{s,\cdot} \\
&\qquad - \frac{\sum_a Q_{t,\lambda}^{\pi_\theta}(s,a)/\lambda - \theta_{s,a}}{|\mathcal{A}|} \Big\|_\infty^2 \\
&\geq \lambda^2 C_\lambda^{\pi_\theta}(s_0) \sum_{s \in \mathcal{S}(s_0)} \sum_{t=0}^{H-1} \mathbb{P}_t^{\pi_\lambda^*}(s|s_0) \Big\| Q_{t,\lambda}^{\pi_\theta}(s,\cdot)/\lambda - \theta_{s,\cdot} - \frac{\sum_a Q_{t,\lambda}^{\pi_\theta}(s,a)/\lambda - \theta_{s,a}}{|\mathcal{A}|} \Big\|_\infty^2 \\
&= \lambda^2 C_\lambda^{\pi_\theta}(s_0) \sum_{t=0}^{H-1} \mathbb{E}_{s \sim \mathbb{P}_t^{\pi_\lambda^*}(\cdot|s_0)} \Big\| Q_{t,\lambda}^{\pi_\theta}(s,\cdot)/\lambda - \theta_{s,\cdot} - \frac{\sum_a Q_{t,\lambda}^{\pi_\theta}(s,a)/\lambda - \theta_{s,a}}{|\mathcal{A}|} \Big\|_\infty^2
\end{aligned}
\tag{A.48}
$$

where

$$
C_\lambda^{\pi_\theta}(s_0) = C_d^2 \min_{t,s \in \mathcal{S}(s_0)} \frac{(\mathbb{P}_t^{\pi_\theta}(s) \min_a \pi_\theta(a|s))^2}{\mathbb{P}_t^{\pi_\lambda^*}(s|s_0)}.
$$

Combining equation A.48 and equation A.43 gives

$$
V_\lambda^{\pi_\lambda^*}(s_0) - V_\lambda^{\pi_\theta}(s_0) \leq \frac{1}{2\lambda} \frac{1}{C_\lambda^{\pi_\theta}(s_0)} \|\nabla V_\lambda^{\pi_\theta}(\mathcal{D})\|^2
\tag{A.49}
$$

which completes the proof. □

## A.5 TOY VERIFICATION IN FIGURE 1

To verify the claim made after Proposition 2, we will need to vary the sparsity of optimal tokens and observe experimental results. However, it is generally difficult to control the sparsity of optimal

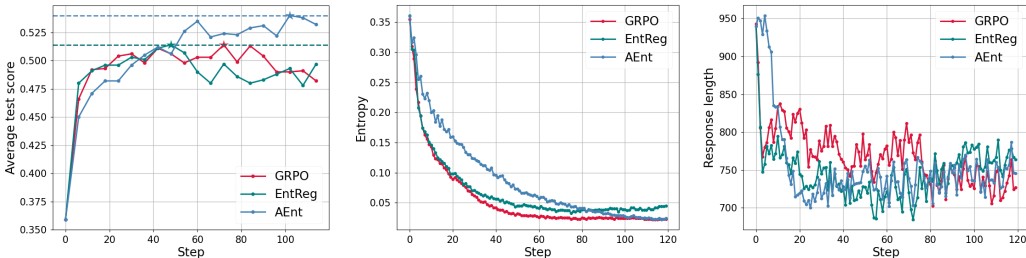

Figure 7: Results of training Qwen2.5-Math-7B on 6k samples from the DeepMath dataset.

Table 3: Benchmark scores of training Qwen2.5-Math-7B on 6k samples from the DeepMath dataset. **Bold** numbers indicate the best result on the benchmark.

|  | MATH-Hard | MATH-500 | AIME24 | Minerva | Olympiad | AMC |
|---|---|---|---|---|---|---|
| **Base** | 0.443 | 0.626 | 0.183 | 0.143 | 0.290 | 0.469 |
| **GRPO** | 0.622 | 0.808 | 0.250 | 0.358 | 0.412 | 0.631 |
| **EntReg** | 0.620 | 0.810 | 0.214 | 0.365 | 0.437 | **0.644** |
| **AEnt** | **0.657** | **0.828** | **0.258** | **0.379** | **0.493** | 0.637 |

responses for real queries. Thus the verification experiments reported in Figure 1 are conducted in a synthetic task, and we leave the results on non-synthetic tasks to Section 5.

**Task setting.** In the synthetic task, the total number of actions is $|\mathcal{A}| = 10^5$, where the number of optimal action $n_{\text{opt}}$ can be picked from $\{15, 10, 5, 1\}$ and the number of suboptimal action is fixed at $500$. The reward for optimal action is $1$, for suboptimal action is $0.2$ and is $0$ for all other actions. For simplicity, we set $H = 1$. The policy is given by a tabular softmax with parameter $\theta \in \mathbb{R}^{10^5}$. To mimic a pre-trained initial policy, we initialize the policy parameter corresponding to the optimal actions and $500$ other random actions from $\mathcal{N}(1, 1)$; while we initialize all other logits from $\mathcal{N}(0, 1)$. Results in the figure are averaged over $20$ independent runs.

**Hyper-parameters.** The learning rate is set as $0.02$, and the batch size is $64$. The hyper-parameters for each algorithm are found through a grid search. For $n_{\text{opt}} = 15, 10, 5, 1$: the regularization coefficient for entropy regularization is set as $0.0005, 0.0005, 0.0005, 0.0007$, and the coefficient for clamped entropy regularization is set as $0.0008$ uniformly. The clamping percentage $p$ is set as $0.98, 0.98, 0.985, 0.997$.

### A.6 ADDITIONAL EXPERIMENTS

In this section, we report some additional experimental results.

**Experimental details.** The algorithms are tested on Qwen2.5-math-7b base model on $6144$ samples from the DeepMath dataset (He et al., 2025). We randomly select queries with over-long filtering (no more than $1024$ tokens) from the dataset. The evaluation method is consistent with that in Section 5.1. We use AdamW optimizer with a learning rate of $1 \times 10^{-5}$. We set the max response length as $3072$. We use a batch size of $512$, and for each query we roll out $8$ responses. For AEnt, we use the GRPO objective as $\mathcal{L}_{\text{PO}}$. We use a clamping percentage $p = 0.33$, and set $\tilde{\mathcal{H}}_{\text{low}} = 0$ and $\tilde{\mathcal{H}}_{\text{high}} = 0.3$. We use an initial entropy coefficient of $0.002$, and start updating the coefficient from the third epoch with $\beta = 0.001$. We clip the coefficient in between $0.0006$ and $0.005$. For EntReg method, we use the traditional entropy bonus with an entropy coefficient of $0.002$.

**Observations.** The results are reported in Figure 7 and Table 3. The performance observation of Figure is overall consistent with that in Section 5.2. It can be observed that AEnt is able to outperform the baselines (left plot) potentially by preventing an early entropy collapsing (middle plot). The response length of the algorithms is overall similar towards the late training period. AEnt's training reasoning efficiency is thus similar to the baselines in this test.

