# OpenReview forum: "On Entropy Control in LLM-RL Algorithms"
_ICLR.cc/2026/Conference — ICLR 2026 Poster_

### Official Review · Reviewer_oda4 · 2025-10-19

**Soundness:** 3
**Presentation:** 3
**Contribution:** 3
**Rating:** 8
**Confidence:** 3

**Summary:**

This paper presents a technique, AEnt, that makes the PPO's (GROP's) entropy regularization work with LLM-RL setting by clamping action-space, focusing on top-N tokens. Previously, the entropy regularization didn't work well in LLM-RL setup due to the extremely large action-space, compared to standard RL setup. In AEnt, this issue is mitigated by clamping the action-space and additionally introducing an algorithm that automatically tunes the entropy coefficient. In experiments, AEnt shows its strong performance improvements as well as the sane behavior of the policy entropy. Ablation studies show that the policy entropy cannot behave well without the proposed clamping mechanism and the automatic coefficienct tuning.

**Strengths:**

This paper has strengths as follows:
- The idea introduced in AEnt is simple and generic enough to be applied to general LLM-RL algorithms.
- In the main experiment, AEnt is shown to be improving performance from the baselines.
- In ablation studies, the changes introduced in AEnt is vital to gain performance improvements.

**Weaknesses:**

The paper is well written. Thus, there isn't obvious flaws in the manuscript. However, here is a list of comments to improve the quality of this paper.
- In the context of automatic entropy cofficient tuning, SAC has also proposed the similar mechanism that is implemented as a dual optimization problem. Authors should explain how the proposed tuning algorithm is selected over SAC's algorithm.
- In Figure 4, the response length plots aren't really described in the body texts. Since you plot them along with the entropy curves, authors should add some descriptions about correlation between the entropy and the response length at least.
- Authors should explain how many random seeds each does each experiment run with.

**Questions:**

The weakness includes my questions that I want to clarify.

---

> ### Author Response · Authors · 2025-11-25
> **Response to Reviewer oda4**
>
> Thank you for the careful review and support. Our response to your comments follows.
>
> **1. In the context of automatic entropy cofficient tuning, SAC has also proposed the similar mechanism that is implemented as a dual optimization problem. Authors should explain how the proposed tuning algorithm is selected over SAC's algorithm.**
>
> According to the Appendix D in SAC [1], it manually adjusts the reward scale coefficient which is essentially the inverse of entropy coefficient. We are somewhat unsure about the exact formulation of the dual optimization problem referred in the comment. If we have misunderstood something, please feel free to elaborate the question further.
>
>
> **2. In Figure 4, the response length plots aren't really described in the body texts. Since you plot them along with the entropy curves, authors should add some descriptions about correlation between the entropy and the response length at least.**
>
> Thank you for the valuable suggestion. We originally included the entropy plot to give insight into the algorithms' entropy trends discussed in  Section 5.2, and we included the response length plots to provide information on algorithm's generation efficiency. With this comment, we think it is beneficial to include additional discussion on these metrics. Thus we have added the following discussion in Section 5.2:
>
> Additionally, it can be observed from Figure 4 that the entropy regularization methods result in slightly longer response in the mid/end of the training period. The potential reason is that the regularization makes the model less certain, and thus the models tend to continue its generation, resulting in longer response. Nonetheless, the increase in response length is relatively mild and we did not observe a major drawback in reasoning efficiency.
>
> **3. Authors should explain how many random seeds each does each experiment run with.**
>
> The results are averaged on two independent runs. We observed that the variance in highest test score is low potentially due to the scores are averaged over multiple benchmarks. To reduce overall observation noise, we tested across three settings with different models and training datasets, with one of the settings done in the rebuttal period (see Appendix A.6 for the added expriment).
>
> Thank you again for the valuable feedback and the support. We sincerely hope our response addresses your questions.
>
> **Reference**
>
> [1] Haarnoja, Tuomas, et al. "Soft actor-critic: Off-policy maximum entropy deep reinforcement learning with a stochastic actor." International conference on machine learning. Pmlr, 2018.

---

### Official Review · Reviewer_j9T9 · 2025-10-28

**Soundness:** 3
**Presentation:** 4
**Contribution:** 3
**Rating:** 8
**Confidence:** 4

**Summary:**

The author reveal why entropy regularization  no gains in LLM-RL training.
Through theoritic anysis, the author reveal that entropy will improve the convergence to optimal solution but bring bias.
The bias is closely related to the scale of action space and the sparsity of the optimal outputs.
Based on this anaysis, the proposed method calculated entropy only in a smaller(top 1-p) action space.
experiments in math-reasoning tasks shows that the proposed method outperforms the baselines.

**Strengths:**

I believe this is a strong paper with novel contributions and valuable insights for the LLM-RL community, particularly in the context of entropy tricks. I recommend accepting this paper.

1. The motivation for the work is clearly presented.
2.The theoretical analysis of entropy is well-executed and convincingly explains why the entropy trick cannot be applied directly in LLM-RL.
3.Simple Yet Efficient Method: The proposed method is both straightforward and effective.

**Weaknesses:**

I have a few questions regarding some details, and I believe additional clarity would be helpful for the broader application of this work.

1. **Choice of Top (1 - p) Percent Tokens:**
   Why do you use the top (1 - p) percent of tokens in \( \pi \), rather than applying a threshold(e.g. \pi > \epsilon) to all tokens?

2. **Experimental Results in Figure 3:**
   As shown in Figure 3, the experimental results indicate that the performance is similar during the first 90% of the training steps, with significant improvement occurring only in the last few epochs.
   How does this happen? Could you provide a more detailed curve with additional steps to better illustrate its stability? An explanation would be greatly appreciated.

3. **Related Work in Entropy Analysis and Action Space Shaping/Pruning:**
   Are there existing works that analyze entropy in large or continuous action spaces within traditional RL methods?
   Additionally, is there prior work on action space shaping (continuous) or action space pruning (discrete) in RL methods (e.g., [1], [2], [3], [4])?
   Could these methods be beneficial for applying the entropy trick in LLM-RL?

**References:**

[1] Zhong D, Yang Y, Zhao Q. No Prior Mask: Eliminate Redundant Action for Deep Reinforcement Learning[C]//Proceedings of the AAAI Conference on Artificial Intelligence. 2024, 38(15): 17078-17086.

[2] Baram, N.; Tennenholtz, G.; and Mannor, S. 2021. Action redundancy in reinforcement learning. In Uncertainty in Artifcial Intelligence, 376–385. PMLR.

[3] Tennenholtz, G.; and Mannor, S. 2019. The Natural Language of Actions. ArXiv, abs/1902.01119.

[4] Tavakoli, A.; Pardo, F.; and Kormushev, P. 2018. Action branching architectures for deep reinforcement learning. In Proceedings of the aaai conference on artifcial intelligence, volume 32.

**Questions:**

See above

---

> ### Author Response · Authors · 2025-11-25
> **Response to Reviewer j9T9**
>
> Thank you for the valuable feedback and support. Our response to your comments follows.
>
> **1. Choice of Top (1 - p) Percent Tokens: Why do you use the top (1 - p) percent of tokens in ( \pi ), rather than applying a threshold(e.g. \pi > \epsilon) to all tokens?**
>
> The output probability scale of the LLM is sensitive to the input sequence and generally unpredictable during training, and thus it is hard to set a fixed threshold $\epsilon$. For certain input sequence $s$ where the LLM is uncertain, the probabilities for each token can be very small, using $\mathcal{A}(s)=\\{a:\pi(a|s)>\epsilon\\}$ may result in an empty token set $\mathcal{A}(s)$ and thus the clamped entropy cannot be calculated. While setting a smaller $\epsilon$ may result in an almost full token set and reduce the regularizer to vanilla entropy. This issue can be avoided by using the $1-p$ percent tokens which is agnostic to the output probability scale of the LLM.
>
> **2. Experimental Results in Figure 3: As shown in Figure 3, the experimental results indicate that the performance is similar during the first 90% of the training steps, with significant improvement occurring only in the last few epochs. How does this happen? Could you provide a more detailed curve with additional steps to better illustrate its stability? An explanation would be greatly appreciated.**
>
> A frequently observed dynamic of RL training without entropy regularization is: as the actions which get sampled and receive higher rewards than other sampled actions get reinforced, the policy becomes more concentrated on these actions and the randomness (entropy) drops. When the entropy drops to zero, the policy becomes stationary and its reward ceases to increase. We can observe this effect by checking Figure 3.(a) (left) paired with Figure 4.(a) (left) where until step 150, GRPO without entropy regularization (red curve) consumes entropy as test reward increases. After step 150 when the entropy decreased to almost zero, the policy becomes stationary and its performance plateaus. On the other hand, AEnt (blue curve) is able to prevent early entropy collapse, which keeps the policy from converging prematurely. It can be observed that the policy continues to imporve after that, ultimately surpassing the baselines.
>
> **3.Are there existing works that analyze entropy in large or continuous action spaces within traditional RL methods? Additionally, is there prior work on action space shaping (continuous) or action space pruning (discrete) in RL methods (e.g., [1], [2], [3], [4])?
> Could these methods be beneficial for applying the entropy trick in LLM-RL?**
>
> Thank you for providing the valuable references. After carefully reading through the listed papers, we believe they provide valuable ideas towards designing LLM entropy regularizers. We have added the following discussion in Section 6 of the main paper:
>
> In addition, we believe the choice of the clamped space $\mathcal{A}(s)$ is crucial to the algorithm's effectiveness, and finding a better choice can potentially yield significant performance gains. For example, one may consider removing actions that are redundant [1,2] or grouping similar actions in entropy calculation. Alternatively, since computing entropy on token level suffers from the large dimensionality of LLM vocabulary, it can be beneficial to design entropy regularizer in the state or action representation space [3,4].
>
> Thank you again for the valuable suggestions and the support for this work. Hope this response addresses your concerns.
>
> **Reference**
>
> [1] Zhong D, Yang Y, Zhao Q. No Prior Mask: Eliminate Redundant Action for Deep Reinforcement Learning[C]//Proceedings of the AAAI Conference on Artificial Intelligence. 2024, 38(15): 17078-17086.
>
> [2] Baram, N.; Tennenholtz, G.; and Mannor, S. 2021. Action redundancy in reinforcement learning. In Uncertainty in Artifcial Intelligence, 376–385. PMLR.
>
> [3] Tennenholtz, G.; and Mannor, S. 2019. The Natural Language of Actions. ArXiv, abs/1902.01119.
>
> [4] Tavakoli, A.; Pardo, F.; and Kormushev, P. 2018. Action branching architectures for deep reinforcement learning. In Proceedings of the aaai conference on artifcial intelligence, volume 32.

---

### Official Review · Reviewer_3biZ · 2025-11-01

**Soundness:** 3
**Presentation:** 3
**Contribution:** 3
**Rating:** 6
**Confidence:** 4

**Summary:**

This paper addresses the limited effectiveness of conventional entropy regularization in LLM-RL training, which contrasts sharply with its success in robotics and games. The authors provide theoretical analysis showing that entropy regularization suffers from bias that scales with response space size and optimal action sparsity. They propose AEnt, which employs two key mechanisms: (1) clamped entropy computed on a re-normalized policy over a reduced token space containing only the top probability tokens, and (2) an adaptive coefficient that automatically adjusts based on whether current entropy falls within target bounds. The method is evaluated on mathematical reasoning tasks using 1.5B parameter models (Qwen2.5-math-1.5b and DeepSeek-R1-distilled-Qwen-1.5b), demonstrating improvements over GRPO and standard entropy regularization baselines on 5 out of 6 benchmarks across two experimental settings.

**Strengths:**

- The paper provides clear theoretical motivation through Propositions 1 and 2, explaining why conventional entropy regularization underperforms in LLM settings due to large action spaces and sparse optimality, with rigorous proofs and synthetic validation in Figure 1.
- The proposed clamped entropy mechanism directly addresses the identified theoretical limitation by restricting entropy maximization to a compact, relevant token space, reducing bias from irrelevant tokens while maintaining exploration benefits.
- Experimental validation includes relevant ablation studies demonstrating the necessity of both adaptive coefficients (Figure 5) and appropriate clamping percentages (Figure 6), with consistent improvements across multiple mathematical reasoning benchmarks.

**Weaknesses:**

- Experimental validation is narrowly confined to mathematical reasoning with 1.5B models, providing insufficient evidence for a method claiming to solve a fundamental LLM-RL problem. Mathematical reasoning has unique properties (verifiable correctness, structured outputs) that may not generalize to open-ended generation, instruction following, or creative tasks where optimal response sets are less well-defined.
- The adaptive coefficient scheme essentially performs online hyperparameter tuning during training, introducing additional hyperparameters (H̃_low, H̃_high, β, λ_low, λ_high) that require careful task-specific tuning. This increases rather than decreases the complexity of deployment, contradicting the implicit promise of a more robust entropy control method.

**Questions:**

- Have you evaluated the method on larger models (7B, 13B, 70B parameters)? The experiments exclusively use 1.5B models, but larger models may exhibit fundamentally different entropy dynamics due to better pre-training, sharper probability distributions, or different exploration-exploitation trade-offs. Specifically, do larger models already concentrate probability mass appropriately, making clamped entropy redundant? Additionally, how does the computational overhead of computing clamped entropy scale with model size, and at what scale does this become prohibitive relative to the performance gains observed?
- Why specifically use top-probability tokens for defining the clamped space, and have you systematically compared against alternative criteria? The choice appears to assume pre-trained models already assign high probability to reasonable tokens, which may be circular reasoning if the goal is to improve exploration. Have you experimented with task-relevant vocabularies defined by domain experts or training data or learned token importance via a separate lightweight model? Furthermore, does the top-p criterion work equally well across different input contexts—for instance, does it fail when the model is uncertain and distributes probability broadly, or when pre-training biases are strong but incorrect for the RL task?
- How does performance vary across fundamentally different task domains beyond mathematical reasoning? Mathematical reasoning has unique properties—verifiable correctness, structured formats, relatively sparse optimal solution sets—that may make it particularly amenable to entropy control. Have you tested on tasks such as open-ended instruction following where response diversity is valued or multi-turn dialogues which require contextual adaptation? More critically, does your theoretical analysis suggesting sparse optimality as problematic even apply to domains where optimal response sets are large or poorly defined? Without broader evaluation, it remains unclear whether you have solved a mathematical reasoning-specific problem or a general LLM-RL challenge.

---

> ### Author Response · Authors · 2025-11-25
> **Response to Reviewer 3biZ (Part 1/3)**
>
> Thank you for the careful review and support. Our response to your comments follows.
>
> **1. Experimental validation is narrowly confined to mathematical reasoning with 1.5B models, providing insufficient evidence for a method claiming to solve a fundamental LLM-RL problem. Mathematical reasoning has unique properties (verifiable correctness, structured outputs) that may not generalize to open-ended generation, instruction following, or creative tasks where optimal response sets are less well-defined.**
>
> We agree that it is of great value to perform RL on broad and open-ended generation tasks. However, we beileve training and evaluation on verifiable tasks provides reliable and useful insight into algorithm performance, and RL on broad domain tasks may still face some foundamental obstacles.  Mathematical tasks require the LLM to perform hierarchial actions (e.g., self-verification, step-by-step reasoning, etc), which is a suitable goal for RL training. Also since the response correctness is verifiable, the reward signals and especially the evaluation results have low variance/bias and are generally reliable. On the other hand, to perform RL on open-ended tasks like creative writing, people often uses LLM-judges or reward models to provide reward signals and evaluation results. However, these model-based rewards/evaluators are highly gameable due to model errors (see, e.g., a review on reward hacking issue in LLM setting [1]), and are susceptible to model bias. Due to these obstacles, it might be hard to fairly compare RL algorithms on these tasks. We dedicate these efforts to future works.
>
>
> **2. The adaptive coefficient scheme essentially performs online hyperparameter tuning during training, introducing additional hyperparameters ($\tilde{H}\_{low}$, $\tilde{H}\_{high}$, $\beta$, $\lambda\_{low}$, $\lambda\_{high}$) that require careful task-specific tuning. This increases rather than decreases the complexity of deployment, contradicting the implicit promise of a more robust entropy control method.**
>
> In our experiments, we did not carefully tune the adaptive coefficient hyper-parameters ($\tilde{H}\_{low}$, $\tilde{H}\_{high}$, $\beta$, $\lambda\_{low}$, $\lambda\_{high}$). We generally used a heuristic for different models and datasets, e.g., $\tilde{H}\_{high}\approx 0.5*H\_0$ when the initial entropy $H\_0$ is high, and a small enough $\beta \leq \lambda$, etc. We find these hyper-paramters are enough to adjust the entropy coefficient to prevent pitfalls as reported in Figure 5. We hypothesize that adaptive coefficient hyper-parameters loosely found from empirical evidences are sufficient for stable training.

---

> ### Author Response · Authors · 2025-11-25
> **Response to Reviewer 3biZ (Part 2/3)**
>
> **3. Have you evaluated the method on larger models (7B, 13B, 70B parameters)? The experiments exclusively use 1.5B models, but larger models may exhibit fundamentally different entropy dynamics due to better pre-training, sharper probability distributions, or different exploration-exploitation trade-offs. Specifically, do larger models already concentrate probability mass appropriately, making clamped entropy redundant? Additionally, how does the computational overhead of computing clamped entropy scale with model size, and at what scale does this become prohibitive relative to the performance gains observed?**
>
> Thank you for the valuable suggestion. Within the limited time of rebuttal period, we have conducted an additional experiment on the larger Qwen2.5-math-7b model and 6k samples from the DeepMath dataset [2]. We report the benchmark scores here.
>
> | | MATH-Hard | MATH-500 | AIME24 | Minerva | Olympiad | AMC |
> | :--- | :--- | :--- | :--- | :--- | :--- | :--- |
> | **Base** | 0.443 | 0.626 | 0.183 | 0.143 | 0.290 | 0.469 |
> | **GRPO** | 0.622 | 0.808 | 0.250 | 0.358 | 0.412 | 0.631 |
> | **EntReg** | 0.620 | 0.810 | 0.214 | 0.365 | 0.437 | **0.644** |
> | **AEnt** | **0.657** | **0.828** | **0.258** | **0.379** | **0.493** | 0.637 |
>
> It can be observed that AEnt outperforms the baselines in five of the six benchmarks. For more training details, please see Appendix A.6 in the revised paper.
>
> Next we answer the more specific questions. Regarding "*do larger models already concentrate probability mass appropriately, making clamped entropy redundant?* ", we believe the benefit of clamped entropy still holds for larger LLMs. Proposition 2 suggests that the entropy bias is increasing with the response space size and decreasing with the optimal response space size. These variables are not dependent on mode size necessarily, and thus larger models still suffer from entropy regularization bias. In this sense, the issue of entropy regularization in LLM tasks still exists, and the intuition of clamped entropy described in Section 4.1 still holds. Empirically, it can be observed from the table above that AEnt's advantage still exsits when tested on larger models.
>
> Regarding "*Additionally, how does the computational overhead of computing clamped entropy scale with model size, and at what scale does this become prohibitive relative to the performance gains observed?* ": In the implementation, the memory overhead of clamped entropy is minimal, and thus we mainly consider the training time complexity. A comparison of total GPU hours required to achieve certain test score is reported in Table 2. Despite clamped entropy takes more computation time each gradient step, it actually takes less total time to achieve GRPO's best score than other baselines. This is due to AEnt needs fewer steps to reach the score threshold, and thus is more sample efficient. Thus in the sense of surpassing the baselines, the proposed method should not incur computation overhead issue. After AEnt surpasses the baseline, it continues to consume compute until it reaches its own best score. We expect the computation time required for this period to scale with 1) the inference time of the model which depends on the model size and the average response length; 2) the size of the training dataset and how informative it is for the model. The precise relationship can only be tested with controlled experiments. Due to the time and computation resource limitation, we could not provide a thorough analysis in the rebuttal priod. Nonetheless, we are definitely insterested in the scaling of AEnt and will report more results in the camera-ready/future version.

---

> ### Author Response · Authors · 2025-11-25
> **Response to Reviewer 3biZ (Part 3/3)**
>
> **4. How does performance vary across fundamentally different task domains beyond mathematical reasoning? Mathematical reasoning has unique properties—verifiable correctness, structured formats, relatively sparse optimal solution sets—that may make it particularly amenable to entropy control. Have you tested on tasks such as open-ended instruction following where response diversity is valued or multi-turn dialogues which require contextual adaptation? More critically, does your theoretical analysis suggesting sparse optimality as problematic even apply to domains where optimal response sets are large or poorly defined? Without broader evaluation, it remains unclear whether you have solved a mathematical reasoning-specific problem or a general LLM-RL challenge.**
>
> Regarding the question about LLM RL on broad domains, we kindly refer the reader to qeustion 1's response.
>
> Regarding "*More critically, does your theoretical analysis suggesting sparse optimality as problematic even apply to domains where optimal response sets are large or poorly defined?* ", we argue that the optimal response set is well defined as long as the LLM task can be formulated as an RL problem, and we believe this set cannot be very large for challenging LLM tasks. Assuming an LLM task can be formulated as an RL problem, the reward function should be properly defined (e.g., correctness in math, human preference reward in creative writing, etc.). Given this reward, the optimal response set $\mathcal{A}_H^*$ is always well defined as in Proposition 1. Furthermore, we believe the optimal responses should still be sparse for challenging queries. For example, in general QA tasks, the optimal responses should at least be semantically meaningful, relevant to the query and on top of that correct. Though hard to quantify, intuitively this set should be much smaller than the complete response set since the complete space consists of large portion of irrelevant and incorrect responses. As the query becomes more challenging, we generally expect the optimal set to shrink.
>
> Thank you again for the valuable feedback and the support for our work. We sincerely hope our response adequately addresses your concerns.
>
> **Reference**
>
> [1] Weng, Lilian. “Reward Hacking in Reinforcement Learning”. 2024. https://lilianweng.github.io/posts/2024-11-28-reward-hacking/.
>
> [2] He, Zhiwei, et al. "Deepmath-103k: A large-scale, challenging, decontaminated, and verifiable mathematical dataset for advancing reasoning." arXiv preprint arXiv:2504.11456. 2025.

---

### Official Review · Reviewer_gqdk · 2025-11-01

**Soundness:** 2
**Presentation:** 3
**Contribution:** 2
**Rating:** 4
**Confidence:** 4

**Summary:**

This paper addresses the well-known problem of standard entropy regularization failing to provide benefits in LLM-RL training, unlike in traditional robotics or game environments. The authors provide a theoretical diagnosis, arguing this failure stems from the combination of the LLM's extremely large response space and the sparsity of optimal outputs, which introduces a significant entropy-induced bias. As a remedy, they propose AEnt, a straightforward entropy control method featuring two key components: 1) a "clamped entropy" bonus calculated only over a size-reduced, high-probability token subspace to mitigate this bias , and 2) an adaptive coefficient that dynamically adjusts to keep this clamped entropy within a target range, preventing training instability. The authors present empirical results on math-reasoning tasks to demonstrate that AEnt can stabilize training and outperform baselines like GRPO and traditional entropy regularization.

**Strengths:**

- The paper studies an important problem that why entropy regularization works for games or robotics but not for llm
- The motivation is strong; the reason why entropy regularization fails in LLMs—attributing it to the large and sparse action space—is intuitive and well-supported. This claim is convincingly backed by both the theoretical bound in Proposition 2 (which explicitly shows the bias term's dependency on action space size $|\mathcal{A}|$ and optimal response sparsity) and a compelling, clear toy experiment in the Figure.

**Weaknesses:**

- The paper's first method is 'clamped entropy' , which means maximizing entropy within the LLM's 'top-p' token range. In fact, this idea is quite straightforward; I have even tried it myself. However, this method is extremely sensitive to the hyperparameter, specifically the coefficient in front of the entropy loss. A good result can be obtained through extensive hyperparameter tuning, but it still requires re-tuning for a new model, and the performance is not very stable.
- The paper's second method, dynamically adjusting the entropy coefficient to maintain stable entropy, is also not a new concept. This adaptive mechanism is functionally similar to early KL control approaches (e.g., [https://arxiv.org/pdf/1909.08593.pdf]) designed to keep the KL divergence within a stable range, a method also integrated into the VeRL repository ([https://github.com/volcengine/verl/blob/main/verl/trainer/ppo/core_algos.py#L150]). To my knowledge, practitioners today still prefer fixed KL coefficients precisely because these dynamic adjustment mechanisms are notoriously unstable. The method's core premise—that forcing entropy to stabilize within a fixed numerical range is beneficial—is unfounded. It is difficult to determine a priori what a "good" range for entropy or KL should be. Furthermore, it is questionable why entropy must be stable; a beneficial training trajectory might just as well involve entropy slowly rising and then slowly falling. The optimal shape of the entropy curve is likely case-by-case, and attempting to manually prescribe its behavior is a questionable engineering choice

- The paper's experimental results are not convincing.
    - In Figure 3, using the peak of the curve to represent the performance is flawed, as it could very likely be an artifact of training randomness.
    - Table 1 only shows the final results for only one model,  but Figure 3 shows the training curves for DeepSeek-R1-distilled-Qwen-1.5b and    Qwen2.5-math-1.5b. Table 1 should be aligned with Figure 3.
    - I would expect the paper to demonstrate consistent improvements on at least three different models without requiring heavy hyperparameter tuning to truly prove the method's effectiveness.

**Questions:**

See weakness

---

> ### Author Response · Authors · 2025-11-25
> **Response to Reviewer gqdk (Part 1/2)**
>
> Thank you for the valuable feedback. Our response to your comments follows.
>
>
> **1. The paper's first method is 'clamped entropy' , which means maximizing entropy within the LLM's 'top-p' token range. In fact, this idea is quite straightforward; I have even tried it myself. However, this method is extremely sensitive to the hyperparameter, specifically the coefficient in front of the entropy loss. A good result can be obtained through extensive hyperparameter tuning, but it still requires re-tuning for a new model, and the performance is not very stable.**
>
> We are sorry for the confusion. We would like to clarify that the proposed 'clamped entropy' does not maximize the entropy on the 'top-p' token range. Instead, it maximizes the entropy on the 'top-k' token range with $k=(1-p)|\mathcal{A}|$. The two methods result in different token set, i.e. different $\mathcal{A}(s)$ in the definition of the clamped entropy $\tilde{H}(\pi_\theta)$, and yield different results. As a result, the observations in this comment might not apply to the proposed method.
>
> Regarding the algorithm's sensitivity to hyper-paramters, our ablation study suggests that although AEnt's performance varies for different hyper-parameters, it still outperforms the baselines on a relative wide range of its unique hyper-parameters.  Specifically, we find that AEnt's performance is mostly sensitive to the entropy coefficient and the clamping percentage $p$, where the entropy coefficient is commonly needed for regularization-type methods including the vanilla entropy regularization method. AEnt's performance is somewhat robust to its unique hyper-parameter clamping percentage $p$ in our experiments. This is observed in Figure 6, where we test $p\in\\{0.2,0.3,0.4\\}$ and observe that all the choices yield advantages over the baselines. In practice, tuning of $p$ is also somewhat intuitive as suggested by the analysis in Section 5.3: one may start from a smaller $p$ and try to increase it until the performance gain drops. This searching rule makes the algorithm easier to tune as compared to when using, e.g., the random/grid search method.
>
> **2. The paper's second method, dynamically adjusting the entropy coefficient to maintain stable entropy, is also not a new concept. This adaptive mechanism is functionally similar to early KL control approaches designed to keep the KL divergence within a stable range, a method also integrated into the VeRL repository...The method's core premise—that forcing entropy to stabilize within a fixed numerical range is beneficial—is unfounded. It is difficult to determine a priori what a "good" range for entropy or KL should be. Furthermore, it is questionable why entropy must be stable; a beneficial training trajectory might just as well involve entropy slowly rising and then slowly falling. The optimal shape of the entropy curve is likely case-by-case, and attempting to manually prescribe its behavior is a questionable engineering choice.**
>
> We would like to clarify that the proposed adaptive coefficient method is not designed to forcefully stabilize entropy. Instead, the method uses a range $ [\tilde{H}\_{low},\tilde{H}\_{high}]$ where the entropy is loosely confined in. Policy entropy can change naturally within the range. The goal is to prevent unstable training phenomenons like entropy/response length exploding observed in our experiments.
>
> In the experiments, it is found out that the adaptive coefficient helps increase sampling efficiency and training stability. It can be observed in Figure 5 that using a constant coefficient results in entropy and response length explosion in the mid of training, while the test accuracy does not benefit from this extra cost in inference. Since the early entropy curve is normal and this issue is unable to be predicted from the early training trend, an adaptive coefficient mechanism is needed to alleviate this issue. It can be observed in Figure 5 that adaptive coefficient is able to prevent the entropy/response length explosion, resulting in significantly better sampling efficiency than constant coefficient.

---

> ### Author Response · Authors · 2025-11-25
> **Response to Reviewer gqdk (Part 2/2)**
>
> **3. In Figure 3, using the peak of the curve to represent the performance is flawed, as it could very likely be an artifact of training randomness.**
>
> Figure 3 demonstrates clear separation of each method's performance, and the peak of the curve is only highlighted to help observe each method's best accuracy. In addition, experiments are conducted on multiple benchmarks and on different models/training datasets. Table 1 and 3 report that AEnt outperforms the baselines on average, and on 5 out of the 6 benchmarks across the three different experimental settings. As a result, AEnt's advantage is unlikely merely an artifact of randomness.
>
> **4. Table 1 only shows the final results for only one model, but Figure 3 shows the training curves for DeepSeek-R1-distilled-Qwen-1.5b and Qwen2.5-math-1.5b. Table 1 should be aligned with Figure 3.**
>
> We are sorry for the confusion. Table 1 has columns "setting (a)" and "setting (b)" which correspond to Qwen2.5-math-1.5b+MATH and DeepSeek-R1-distilled-Qwen-1.5b+OpenR1-Math respectively. The detailed setting is described in Section 5.1.
>
> **5. I would expect the paper to demonstrate consistent improvements on at least three different models without requiring heavy hyperparameter tuning to truly prove the method's effectiveness.**
>
> Thank you for the valuable suggestion. Within the limited time of rebuttal period, we have conducted an additional experiment on training Qwen2.5-math-7b on 6k samples from the DeepMath training dataset [1]. Next we report the benchmark scores.
>
>
> | | MATH-Hard | MATH-500 | AIME24 | Minerva | Olympiad | AMC |
> | :--- | :--- | :--- | :--- | :--- | :--- | :--- |
> | **Base** | 0.443 | 0.626 | 0.183 | 0.143 | 0.290 | 0.469 |
> | **GRPO** | 0.622 | 0.808 | 0.250 | 0.358 | 0.412 | 0.631 |
> | **EntReg** | 0.620 | 0.810 | 0.214 | 0.365 | 0.437 | **0.644** |
> | **AEnt** | **0.657** | **0.828** | **0.258** | **0.379** | **0.493** | 0.637 |
>
> It can be observed that AEnt outperforms the baselines in five of the six benchmarks with a significant advantage on Olympiadbench. For more training details and analysis, please see Appendix A.6 in the revised paper. For AEnt, we searched the clamping percentage $p$ in $\{0.25,0.3,0.33\}$ and did not tune other AEnt-unique hyper-parameters due to time limitation. As a result, the tuning effort was limited and relatively light-weight.
>
> Thank you again for the careful review. We hope that our response addresses your concerns, potentially leading to a higher score. Please let us know if further clarification is required.
>
> **Reference**
>
> [1] He, Zhiwei, et al. "Deepmath-103k: A large-scale, challenging, decontaminated, and verifiable mathematical dataset for advancing reasoning." arXiv preprint arXiv:2504.11456. 2025.

---

> > ### Comment · Reviewer_gqdk · 2025-11-26
> >
> > I thank the authors for their detailed response. However, after carefully reviewing the rebuttal, my main concerns remain unresolved:
> > - Regarding W1: I am not convinced by the distinction between optimizing top-p tokens versus top-k tokens. In my view, both approaches are straightforward heuristics.
> >
> > - Regarding W2: The issue regarding the entropy constraint is not fully addressed. Constraining entropy within a fixed range relies heavily on two additional hyperparameters (the upper and lower bounds), which appear to be sensitive and require case-by-case tuning.
> >
> > - Regarding W3: While this point was partially addressed, the robustness of the method remains unproven. The dependence on specific hyperparameters—specifically the entropy bounds and the clamping percentage $p$—suggests a lack of generalization.  **I suspect that these parameters would need to be re-tuned for different models, datasets, or even slight changes in system prompts.**
> >
> > Overall, I perceive the proposed method as primarily an engineering trick rather than a theoretically sound algorithmic innovation. Due to the lack of sufficient novelty and robustness, I will maintain my rating (4).

---

> > > ### Author Response · Authors · 2025-12-03
> > >
> > > We sincerely thank the reviewer for the feedback. We are happy to see the added experiments addressed parts of the reviewer's concerns. Though the discussion phase was ended prematurely due to unforeseen circumstances, we still would like to add a final follow-up comment on the reviewer's remaining concern.
> > >
> > > We would like to argue that 1) the issue of entropy bonus under LLM setting is rigorously shown in Proposition 2, which provides grounding and clear intuition for AEnt described in Section 4.1; and, 2) though the algorithm's performance varies for different hyper-parameters, its advantage over the baselines is relatively robust and consistently observed in the experiments. For example, it can be observed in Figure 6 that AEnt consistently outperforms the baselines for $p\in\\{0.2,0.3,0.4\\}$, which is not limited to our choice of $p=0.33$. Additionally, in the added experiment in Appendix A.6, we did not carefully tune AEnt's hyper-parameters due to time limitation of the rebuttal period. In this new experiment setup, we used $p=0.33$ which is the same as that used for a different experiment setup described in line 361. Despite the difference in model size and training dataset, AEnt outperforms the baselines as observed in Figure 7. Supported by these evidences, we believe the algorithm's performance is robust to its unique hyper-parameters, and light-weight tuning is enough for performance gains.
> > >
> > > We would like to thank the reviewer again for the valuable feedback.

---

### Meta-Review · Area_Chair_m3YU · 2025-12-25

**Summary:**

The decision to accept this paper is driven by the authors' successful rebuttal to the reviewers' primary concerns regarding the method's scalability and robustness. While reviewers initially questioned the applicability of the proposed AEnt method to larger models and its theoretical novelty, the authors provided convincing evidence during the discussion. Specifically, the additional experiments on a 7B parameter model directly addressed the critical concern about scalability. Although some concerns regarding generalization to non-mathematical domains and the perception of the method as an "engineering trick" remain, the demonstrated consistent performance improvements on math reasoning tasks and the theoretical grounding provided are sufficient to merit acceptance.

**Reviewer Concerns:**

**Addressed Concerns:**
*   **Evaluation on Larger Models**: Reviewer 3biZ's concern about the limitation to 1.5B models was effectively addressed by new experiments on Qwen2.5-math-7b, showing consistent gains.
*   **Hyperparameter Sensitivity**: Reviewer 3biZ's worry about tuning complexity was addressed by the authors' explanation of robust heuristics and sensitivity analysis.
*   **Technical Details**: Questions from Reviewers j9T9 and oda4 regarding the top-p choice and comparison to SAC were clarified satisfactorily.

**Outstanding Concerns:**
*   **Generalization to Other Domains**: Reviewer 3biZ's question about performance on open-ended tasks (like creative writing) remains a limitation. The authors argued that verifiable tasks are better for fair comparison, leaving broader application to future work.
*   **Novelty vs. Engineering Trick**: Reviewer gqdk's perception of the method as primarily an engineering trick rather than a theoretical innovation persists. While the authors pointed to Proposition 2 for theoretical grounding, this fundamental disagreement on the nature of the contribution likely remains.

**Reviewer Scores:**

*   **Reviewer gqdk**: 4 -> 4. The reviewer might maintain their view on novelty, though the robustness evidence is strong.
*   **Reviewer 3biZ**: 6 -> 8. The primary concern regarding model scale was directly addressed with new experiments on a 7B model, which should significantly strengthen the reviewer's assessment.
*   **Reviewer j9T9**: 8 -> 8. The reviewer was already positive and satisfied with the clarifications.
*   **Reviewer oda4**: 8 -> 8. The reviewer was already positive and satisfied with the clarifications.

---

### Decision · Program_Chairs · 2026-01-26

Accept (Poster)